# Grafting of iPS cell-derived tenocytes promotes motor function recovery after Achilles tendon rupture

Taiki Nakajima [1 ✉], Akihiro Nakahata[2], Naoki Yamada[1,3], Keiko Yoshizawa[1], Tomoaki M. Kato[4], Mio Iwasaki[5], Chengzhu Zhao[1], Hiroshi Kuroki[2] & Makoto Ikeya[1 ✉]

Tendon self-renewal is a rare occurrence because of the poor vascularization of this tissue; therefore, reconstructive surgery using autologous tendon is often performed in severe injury cases. However, the post-surgery re-injury rate is relatively high, and the collection of autologous tendons leads to muscle weakness, resulting in prolonged rehabilitation. Here, we introduce an induced pluripotent stem cell (iPSC)-based technology to develop a therapeutic option for tendon injury. First, we derived tenocytes from human iPSCs by recapitulating the normal progression of step-wise narrowing fate decisions in vertebrate embryos. We used single-cell RNA sequencing to analyze the developmental trajectory of iPSC-derived tenocytes. We demonstrated that iPSC-tenocyte grafting contributed to motor function recovery after Achilles tendon injury in rats via engraftment and paracrine effects. The biomechanical strength of regenerated tendons was comparable to that of healthy tendons. We suggest that iPSC-tenocytes will provide a therapeutic option for tendon injury.

[1] Department of Clinical Application, Center for iPS Cell Research and Application, Kyoto University, Kyoto, Japan. [2] Department of Human Health Sciences, Graduate School of Medicine, Kyoto University, Kyoto, Japan. [3] Department of Rehabilitation Medicine, Jikei University School of Medicine, Tokyo, Japan. [4] Facility for iPS Cell Therapy, CiRA Foundation, Kyoto, Japan. [5] Department of Life Science Frontiers, Center for iPS Cell Research and Application, Kyoto University, Kyoto, Japan. ✉email: nakajima.taiki.84r@kyoto-u.jp; mikeya@cira.kyoto-u.ac.jp

Tendons are soft tissues with remarkably high tensile strength and power that transmit the mechanical forces of muscular contraction to the bones. Sustaining tendon injury severely affects the quality of life, especially for elderly people and athletes; thus, fast recovery is essential. Conventionally, reconstructive surgery using autologous tendon is performed in case of severe injury; however, the risk of complications is high due to the invasive nature of the procedure, and the re-injury rate after surgery is quite high[1]. In addition, the collection of autologous tendon tissue leads to muscle weakness, inevitably prolonging rehabilitation[2]. Although advances have been made in developing artificial tendons made from collagen, carbon, or polyester[3], issues such as low compatibility and bioadhesiveness remain. A promising approach to overcome these issues is cell transplantation; grafting of primary tenocytes, dermal fibroblasts, or mesenchymal stem cells (MSCs) can promote tenocyte recruitment and regeneration of injured tendons[4–10]. However, collecting a large number of primary tenocytes is difficult because of their low proliferative ability, and in vitro expansion can cause phenotypic changes and functional loss[11]. Moreover, grafting of non-tenocytes such as dermal fibroblasts and MSCs carries a risk of ectopic tissue formation as they may produce fibrotic extracellular matrix leading to scar formation, as MSCs can generate bone and cartilage besides tenocytes[11,12].

Therefore, there is a considerable demand for novel technology-based cell therapies for treating tendon injury. We have been developing an induced pluripotent stem cell (iPSC)-based technology and reported the first method for inducing differentiation of iPSC-derived tenocytes (iPSC-tenocytes) that could potentially be applied to transplantation therapy[13–15]. Utilizing terminally differentiated iPSC-tenocytes is promising as it avoids ethical concerns and autologous tissue harvest and because of the high proliferative ability and relatively low risk of ectopic tissue formation[11]. While mouse iPSCs or MSCs have been used to produce tenocytes, the application of human iPSC-derived tenocytes to cell therapy would be innovative[16–23]. One study used embryoid bodies and empirically identified factors that transformed mouse iPSCs to cells that express tenogenic markers such as Scx and Mkx[20]. Another study used lentiviral Scx transgene expression in mouse MSCs to derive tenocytes[22,23]. The grafting of mouse iPSC-derived tenocyte-like cells was shown to reduce scar formation in mice[20], demonstrating the potential of iPSCs in tendon injury treatment; however, many issues are left unsolved. For instance, no therapeutic effect on motor function recovery after transplantation has been proven. Besides, for future medical applications, it is desirable that animal-derived components are excluded from the protocol for human PSC tenocyte differentiation.

Tenocytes have several embryological origins, but in the trunk of the body, they are derived from somites, a family of paraxial mesoderm cells[24]. In this study, we used a development-informed innovative approach based on the understanding of the normal progression of step-wise narrowing fate decisions of tenocytes during embryogenesis to establish a method for efficient differentiation of human iPSCs into tenocytes with characteristics similar to primary tenocytes in animal component-free (xeno-free) conditions. We used Achilles tendon rupture in rats to assess whether transplanted iPSC-tenocytes integrate into host tissues and contribute to motor function recovery.

## Results

### Deriving syndetome from human iPSCs
In the early stage of paraxial mesoderm development, epiblasts located in the anterior primitive streak invaginate inside the embryo and form the presomitic mesoderm (PSM), subsequently forming somites through budding off rostrally[25,26]. Somites adapt their fate in response to signaling molecules secreted from surrounding tissues such as the notochord, neural tube, surface ectoderm, and lateral plate mesoderm, and then subdivide into two compartments: dermomyotome dorsally and sclerotome ventrally[27,28]. The dorsal part of the sclerotome further differentiates into the syndetome, which eventually gives rise to tenocytes in the torso[24,29]. This step-wise narrowing fate decision of the paraxial mesoderm in the embryo can be recapitulated in vitro with PSCs, such as iPSCs or embryonic stem cells[13–15,30–34]. We have previously demonstrated the directed differentiation of tenocytes from human iPSCs by modeling human somite development in vitro[13–15]. Here, we modified our conventional protocols to establish a method to robustly derive tenocytes from human iPSCs in xeno-free conditions using an iPSC line, 1231A3[35].

We cultured the cells in a medium containing the transforming growth factor-β (TGFβ) inhibitor SB431542 (SB), WNT activator CHIR99021 (CHIR), bone morphologic protein (BMP) inhibitor DMH1, and fibroblast growth factor-2 (FGF2) recombinant protein for the first four days, and then in medium containing only SB and CHIR to differentiate the iPSCs into somites through the PSM stage. On day 8, we used a medium containing smoothened agonist (SAG) and the BMP inhibitor LDN193189 (LDN)[36] to differentiate the somites into sclerotomes expressing PAX1. As FGF8 signaling is required for syndetome development in the early phase[24] and BMP and TGFβ signaling are involved in tenocyte development and maintenance[18,37,38], we grew iPSC-derived sclerotome in the presence of FGF8 and TGFβ3 for two days and then in the presence of BMP7 and TGFβ3 for six days (Fig. 1a). A major difference from our previous method is the timing of TGFβ3 treatment as, in the original method, TGFβ3 was administered on day 3 of syndetome differentiation[13]. As shown in Fig. 1b, gene expression of syndetome- and tenocyte-specific transcription factors and extracellular matrix proteins, including SCX, MKX, TNMD, COL1A1, COL1A2, and FMOD, was time-dependently upregulated. On day 19, cells differentiated from iPSCs showed mesenchymal cell-like morphology, resembling primary tenocytes. Immunocytochemistry revealed high protein expression of SCX (91.6%), MKX (90.4%), COL1A1 (79.9%), and COL1A2 (77.5%) (Fig. 1c). To assess the proliferative characteristics of iPSC-syndetome, we then carried out EdU staining and cell growth curve analyses. As shown in Fig. 1d, 6.0% of iPSC-syndetome were EdU-positive, and iPSC-syndetome showed high proliferative properties over 8 days with a stable growth curve until confluence (Supplementary Fig. 1a). These data indicated that our developmentally informed approach provides robust in vitro differentiation of iPSC-syndetome in xeno-free conditions.

### Transcriptional trajectories throughout iPSC-syndetome differentiation
To further characterize the differentiating cells and trace the transcriptional dynamics throughout iPSC-syndetome differentiation, iPSCs, iPSC-derived PSM, somites, sclerotome, and syndetome (days 2, 4, 6, 8) were subjected to single-cell RNA sequencing (scRNA-seq). We first combined data from all time points and then visualized clusters and gene expression on a two-dimensional uniform manifold approximation and projection (UMAP). As shown in Fig. 2a and b, the different cell types were subdivided into distinct clusters that preferentially expressed representative gene markers (iPSCs: NANOG and OCT3/4; PSM: TBX6 and HES7; somites: PAX3 and MEOX1; sclerotome: PAX1 and UNCX; syndetome: SCX and COL1A2). A violin plot showed that the representative gene markers were expressed at each time point (Supplementary Fig. 2a). We next performed heatmap

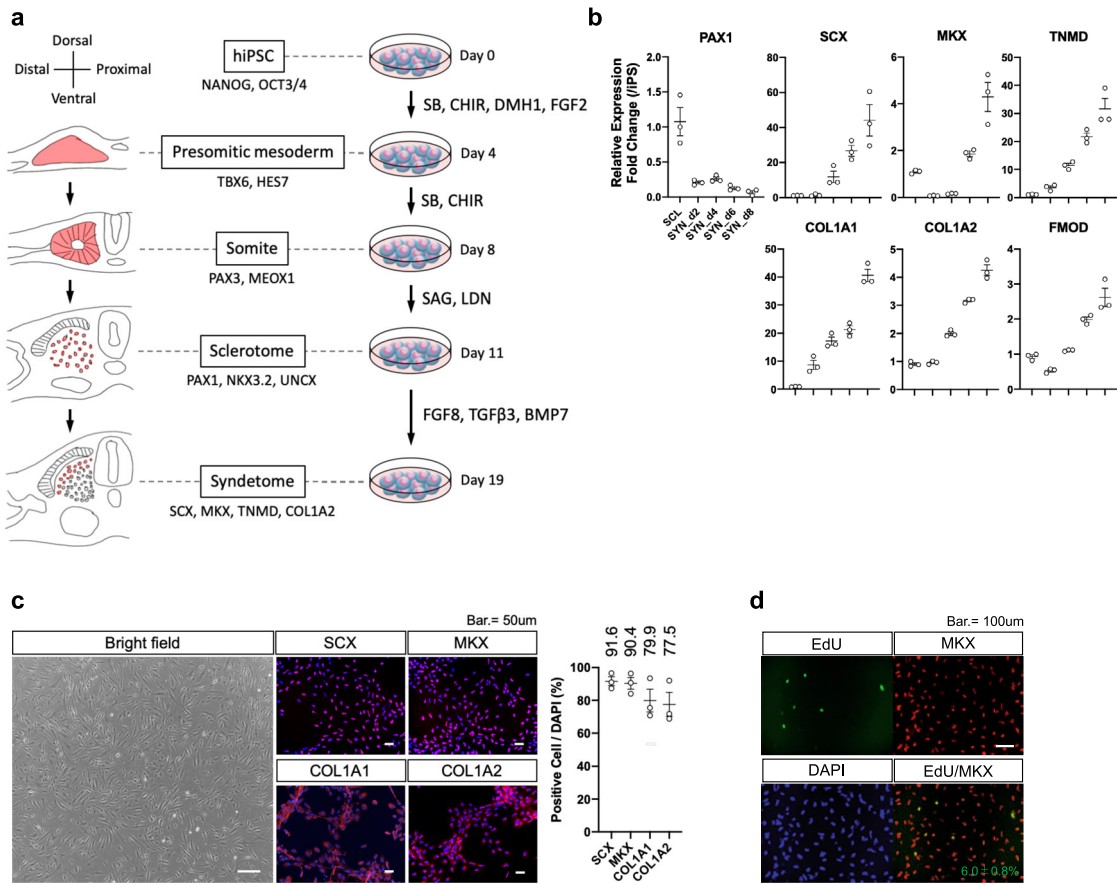

**Fig. 1 Robust differentiation of human iPSC-derived syndetome. a** Modeling step-wise narrowing fate decisions of syndetome in the embryo with human iPSCs. iPSCs were differentiated toward presomitic mesoderm in medium supplemented with SB, CHIR, DMH1, and FGF2 for four days. Subsequently, day-4 presomitic mesoderm cells were passaged and differentiated toward somites in medium supplemented with SB and CHIR for another four days. Day-8 somites were then cultured in a medium containing SAG and LDN for sclerotome differentiation. Day-11 sclerotome cells were passaged and subsequently differentiated toward syndetome in medium supplemented with FGF8 and TGFβ3 for the first two days, and in medium supplemented with TGFβ3 and BMP7 for the following six days. **b, c** Differentiation toward syndetome fate was assessed by RT-qPCR (**b**), bright-field imaging, and immunocytochemistry (**c**). The expression of markers for sclerotome (*PAX1*) and syndetome (*SCX, MKX, TNMD, COL1A1, COL1A2, FMOD*) was assessed by RT-qPCR. Cells were stained with anti-SCX, MKX, COL1A1, COL1A2 antibodies (red) and co-stained with DAPI (blue). Positive rates to DAPI were calculated using Image J. Representative data are shown. Scale bars: 50 μm. **d** EdU staining of iPSC-syndetome. iPSC-syndetome were stained with EdU (green) and co-stained with anti-MKX antibody (red) and DAPI (blue). EdU-positive rates to MKX were calculated using Image J. Representative data are shown. Scale bars: 100 μm. Data represent mean ± SE (*n* = 3: biologically independent samples). iPSC, induced pluripotent stem cell; SB, SB431542; CHIR, CHIR99021; LDN, LDN193189. Source data are provided as a Source Data file.

(Fig. 2c) and dot plot (Fig. 2d and Supplementary Fig. 2b) analyses to identify the most differentially expressed genes at each time point. We thus identified several specific markers at appropriate time points, such as *OCT3/4* (*POU5F1*) in iPSCs, *DLL3* in PSM, *PAX3* in somites, *CRABP1*[39] in sclerotome, and *SCX, COL9A3, COL3A1, COL12A1,* and *POSTN* in syndetome (Fig. 2c). A dot plot of the top 20 genes differentially expressed in syndetome versus sclerotome contained multiple factors known to contribute to tendon healing or tenocyte recruitment (e.g., *POSTN, FIBIN, and SPARC*) and the extracellular matrix of tenocytes (e.g., collagens) (Fig. 2d). These results demonstrated that each cell type exhibits proper transcriptional characteristics throughout iPSC differentiation and that the in vitro differentiation pathway appears to reflect the normal progression of step-wise narrowing fate decisions of the paraxial mesoderm in vertebrate embryos.

**Clustering analysis using iPSC-derived syndetome and primary tenocytes reveals their transcriptional similarities and syndetome homogeneity.** Next, we explored transcriptional similarities

between iPSC-derived syndetome and primary tenocytes. UMAP for iPSC-derived sclerotome, syndetome (days 2, 4, 6, 8), and primary human tenocytes identified 12 unique cell clusters (labeled 0–11) (Fig. 3a). The whole syndetome clusters [0–4, 7] revealed universal expression of representative tenocyte markers such as *SCX, COL1A2, COL3A1,* and *DCN* (Fig. 3b), demonstrating a homogenous syndetome population. As for tenocyte-specific extracellular matrix-related gene expression, syndetome exhibited expression characteristics closer to primary tenocytes than to sclerotome, suggesting a transcriptional similarity between syndetome and primary tenocytes (Fig. 3c). Furthermore, a heatmap showing the top 40 differentially expressed genes in primary human tenocytes vs. iPSC-derived sclerotome exhibited similar expression trends in syndetome and primary tenocytes (Fig. 3d). The color pattern in the heatmap also reflected the gradual differentiation approach from sclerotome toward tenocytes. As Fig. 3a shows, the whole iPSC-derived syndetome (hereafter referred to as iPSC-tenocytes) was divided roughly into two major clusters: cluster [0,1,2,3] and cluster [4,7]. Transcriptional differences were assessed by dot plot analysis

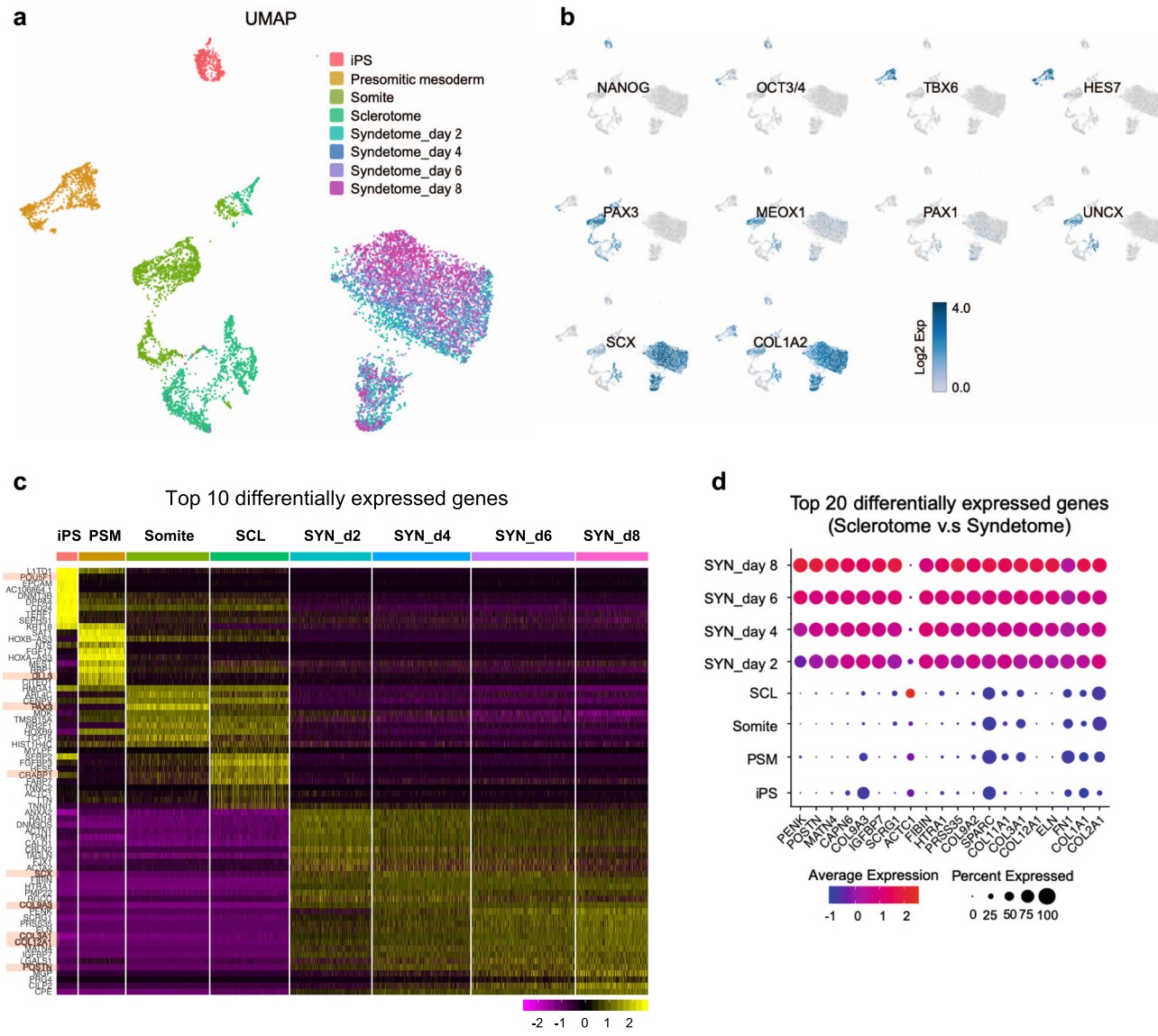

**Fig. 2 Transcriptional dynamics throughout iPSC-syndetome differentiation. a, b** Two-dimensional uniform manifold approximation and projection (UMAP) with combined data from all time points for iPSCs, presomitic mesoderm, somite, sclerotome, and syndetome (**a**). The expression of markers for iPSCs (*NANOG, OCT3/4*), presomitic mesoderm (*TBX6, HES7*), somites (*PAX3, MEOX1*), sclerotome (*PAX1, UNCX*), and syndetome (*SCX, COL1A2*) (**b**). **c** Heatmap of the top 10 differentially expressed genes in each sample. Representative gene markers are labeled. **d** Dot plot of the top 20 differentially expressed genes in sclerotome vs. syndetome. PSM, presomitic mesoderm; SCL, sclerotome; SYN, syndetome.

showing the top 20 differentially expressed genes between the two major clusters. Cell cycle-related genes, including *CENPF*, *TOP2A*, and *MKI67*, accounted for most of the top 20 genes (Fig. 3e). Cell cycle analysis in which cell cycle phase scores were calculated based on canonical markers showed that cluster [0,1,2,3] was in the G1 state, whereas cluster [4,7] covered the G2, M, and S states (Fig. 3f). These data demonstrated that the iPSC-tenocytes were a homogeneous population encompassing various cell cycle phases, whose transcriptional features resembled those of primary tenocytes.

**Transplantation of iPSC-tenocytes contributes to motor function recovery after Achilles tendon rupture in rats.** To assess the therapeutic effects of the iPSC-tenocytes, we established a complete transverse tear on the left Achilles tendons of rats and then transplanted $3 \times 10^6$ iPSC-tenocytes into the injured site using a syringe. The extent of regeneration was evaluated for 4 weeks after surgery by kinematic, biomechanical, and histological testing (Fig. 4a). Footprint analysis showed that healthy rats before surgery did not attach their heels to the floor throughout a sequence of walking actions, whereas rats one week after the operation did, indicating that the Achilles tendon injury diminished motor function. Two weeks after surgery, rats treated with iPSC-tenocytes (iPSC-tenocytes rats) tended to walk with a raised heel, whereas untreated rats (injured, but not transplanted) did not (Fig. 4b, Supplementary Movie 1–4). Using footprint data, we calculated the Achilles functional index (AFI), a marker of motor function recovery after Achilles tendon injury[40], by measuring print length, toe spread, and intermediate toe spread. The AFI score of iPSC-tenocytes rats was significantly higher than that of untreated rats, one and two weeks after transplantation, whereas no significant difference was observed three and four weeks post-surgery (Fig. 4c). To investigate kinematic recovery, we assessed the heel height and ankle angle of rats walking on a treadmill by tracking markers attached to the knee joint, calcaneus, and fifth metatarsal. The heel height of iPSC-tenocytes rats

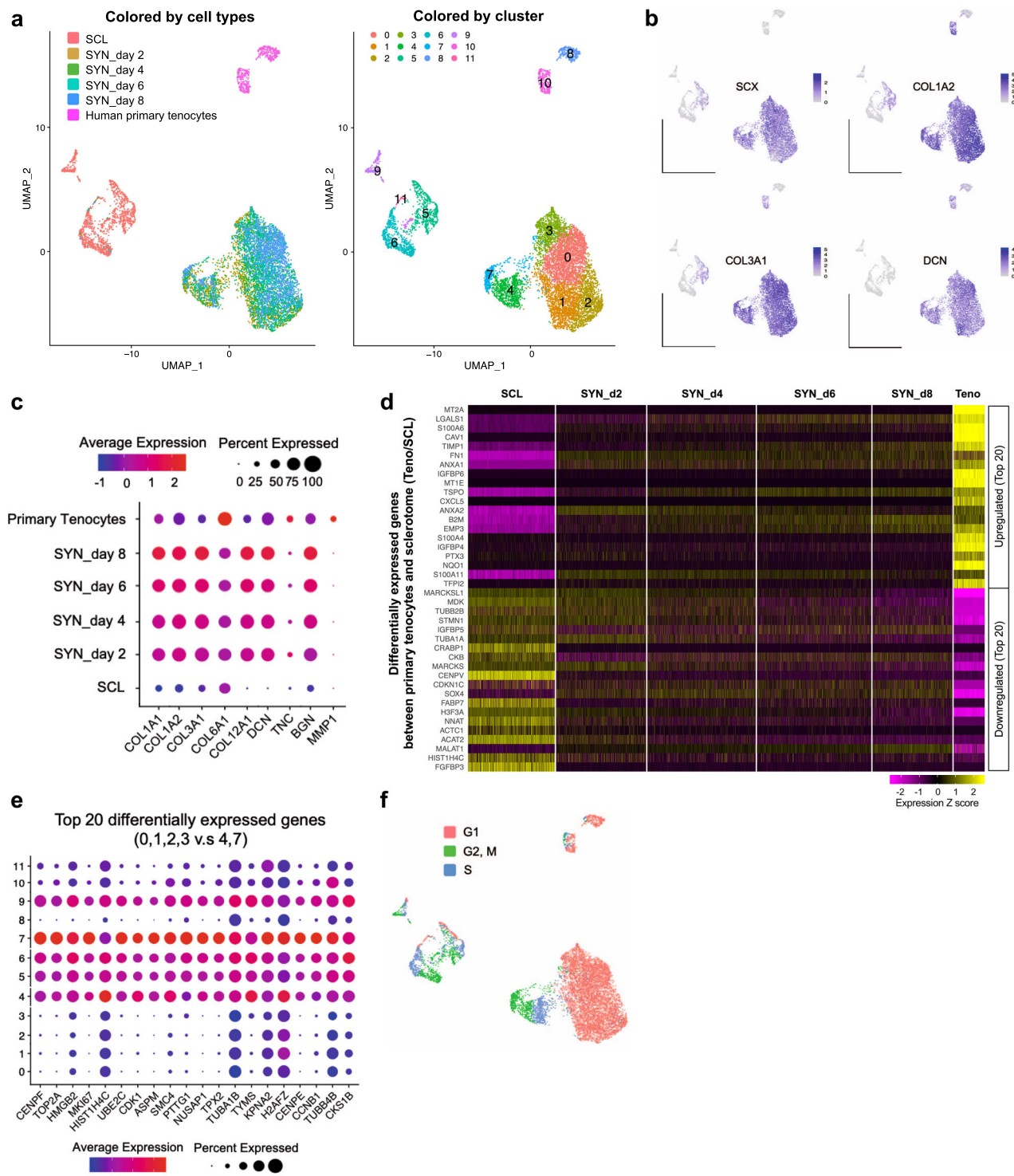

**Fig. 3 Characterization of iPSC-derived syndetome. a, b** UMAP for iPSC-derived sclerotome, syndetome (days 2, 4, 6, 8), and primary human tenocytes revealed 12 unique cell clusters (labeled 0–11) (**a**). Representative markers for syndetome and tenocytes are shown (**b**). **c** Dot plot showing the expression of tenocyte-specific extracellular matrix factors expressed in sclerotome, syndetome, and primary tenocytes. **d** Heatmap showing the top 40 differentially expressed genes in primary human tenocytes vs. iPSC-derived sclerotome. Gene expression in iPSC-derived syndetome indicates the transcriptional similarity of iPSC-derived syndetome and primary human tenocytes. Teno, primary human tenocytes. **e, f** Characteristics of each syndetome cluster were assessed based on a dot plot showing the top 20 differentially expressed genes in each cluster [0,1,2,3] and [4,7] (**e**), and by cell cycle analysis (**f**).

at two weeks was restored and was comparable to that of unin-jured rats (neither incised nor transplanted), but was significantly higher than that of untreated rats (Fig. 4d, left panel). Moreover, the ankle angle of iPSC-tenocytes rats at two and four weeks seemed to be recovered to plantar flexion similar to uninjured

rats. In contrast, untreated rats still showed dorsal flexion even at four weeks (Fig. 4d, right panel), suggesting that the regenerated Achilles tendon of iPSC-tenocytes rats could transmit the output from the gastrocnemius muscle to the bone better than that of untreated rats. These observations imply that regenerated Achilles

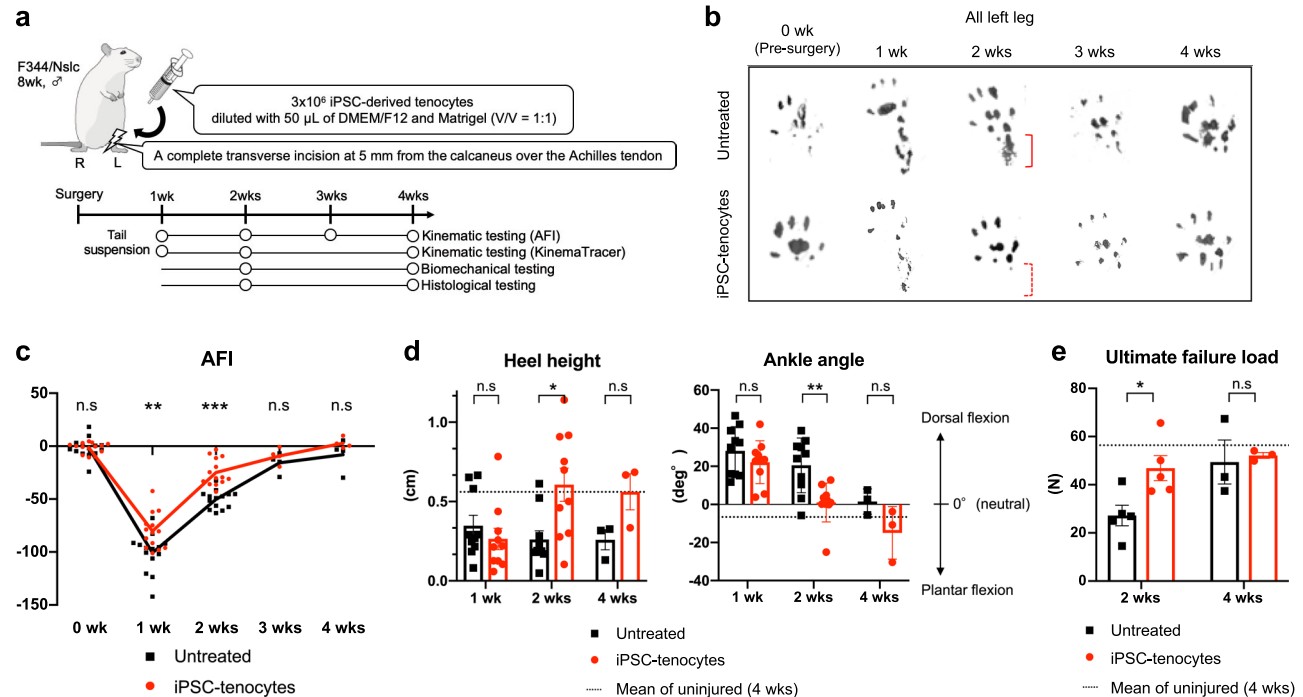

**Fig. 4 Grafting of iPSC-tenocytes promotes the recovery of motor function after Achilles tendon rupture in rats. a** Schematic representation of animal experimental setup. **b** Representative footprints of iPSC-tenocytes rats and untreated rats at weeks 0 (before surgery), 1, 2, 3, and 4 after transplantation. **c**, **d** Kinematic analyses of iPSC-tenocytes rats, untreated rats, and uninjured rats. AFI (Achilles functional index) was measured at weeks 0 (before surgery), 1, 2, 3, and 4 after transplantation. A value of 0 indicates normal function and a negative value indicates impairment (0, 1, 2 weeks: $n = 14$: biologically independent samples; 3 and 4 weeks: $n = 4$: biologically independent samples) (**c**). Heel height (cm) and ankle angle (°) of rats walking on a treadmill were assessed at weeks 1, 2, and 4 after transplantation. A positive angle indicates dorsal flexion, a negative angle indicates plantar flexion (**d**). Data represent mean ± SE (1 and 2 weeks: $n = 10$: biologically independent samples; 4 weeks, uninjured rats: $n = 3$: biologically independent samples). Dotted lines in the figures show the mean score for uninjured rats. **e** Biomechanical analysis of iPSC-tenocytes rats, untreated rats, and uninjured rats. Ultimate failure load (N) of regenerated tendons harvested from rats was assessed at weeks 2 and 4 after transplantation. The dotted line in the figure shows the mean score for uninjured rats. Error bars: mean ± SE (2 weeks: $n = 5$: biologically independent samples; 4 weeks: $n = 3$: biologically independent samples; uninjured rats: $n = 7$: biologically independent samples). $*P < 0.05$; $**P < 0.01$; $***P < 0.001$ by two-tailed Welch's t-test (**c**, **d**, **e**). n.s., not significant; untreated, incised but not transplanted; uninjured, neither incised nor transplanted. Source data are provided as a Source Data file.

tendon treated with iPSC-tenocytes has better durability than untreated tendons; thus, we assessed the ultimate failure load of Achilles tendons collected at two and four weeks. Regenerated tendon treated with iPSC-tenocytes could withstand a higher tensile force at two weeks than untreated Achilles tendon (Fig. 4e). To further characterize the physical properties of the injured area, we analyzed load-displacement curves, which are generally used to judge the properties of substances. Untreated rats showed a gentle curve, which indicates increased ductility and a viscous contribution to the viscoelastic response, whereas iPSC-tenocytes rats and uninjured rats showed similar curves with a sharp drop immediately after the rupture (Supplementary Fig. 3a). In iPSC-tenocytes rats, the fracture load [N] and stiffness [N/mm] at the injured area were recovered compared to those in untreated rats. These observations demonstrated that transplantation of iPSC-tenocytes promoted tendon regeneration and motor function recovery.

To evaluate the cell type specificity of iPSC-tenocytes, we investigated tendon regeneration after iPSC-derived sclerotome transplantation and bone marrow-derived human MSC (BMSC) transplantation. As reported previously[4,5,7], BMSC transplantation contributed to Achilles tendon regeneration, and the therapeutic effect appeared to be comparable to that of iPSC-tenocytes (Supplementary Fig. 4a, b). However, iPSC-derived sclerotome did not significantly affect the AFI (Supplementary Fig. 4a), heel height, and ankle angle (Supplementary Fig. 4b). Alcian blue staining (Supplementary Fig. 4c) and immunohistochemistry using an anti-

SOX9 antibody (Supplementary Fig. 4d) demonstrated that ectopic cartilage tissue developed in the host tendon fibers four weeks after iPSC-derived sclerotome transplantation: two out of two tendons treated with iPSC-sclerotome cells were positive for Alcian blue staining. The fact that iPSC-tenocytes and BMSCs had similar therapeutic effects suggests that they may share common mechanisms in contributing to tendon healing.

**Engrafted iPSC-tenocytes in host tendon fibers express collagen at two weeks after transplantation**. To investigate the mechanism by which the iPSC-tenocytes recovered motor function, we assessed the regeneration from a histological aspect and investigated donor cell engraftment in the host tendon tissue. Hematoxylin and eosin staining and Masson's trichrome staining demonstrated that tendon fibers treated with iPSC-tenocytes appeared well regenerated in terms of cell density (Fig. 5a, b) and fiber alignment at four weeks after transplantation (Fig. 5c). Next, iPSC-tenocytes were pre-stained with a fluorescent dye, Xeno-Light DiR, and the biodistribution of the DiR-labeled cells was monitored at 24 h and 2 and 4 weeks after transplantation. Fluorescence was detected specifically in the grafted area at all three time points (Fig. 5d). We collected regenerated Achilles tendons at two weeks and immunostained them for human-specific vimentin, collagen type I, and collagen type III. Cells expressing vimentin were located in the host tendon fibers, and some of them also expressed collagen type I and collagen type III (Fig. 5e), suggesting that, to some extent, engrafted iPSC-

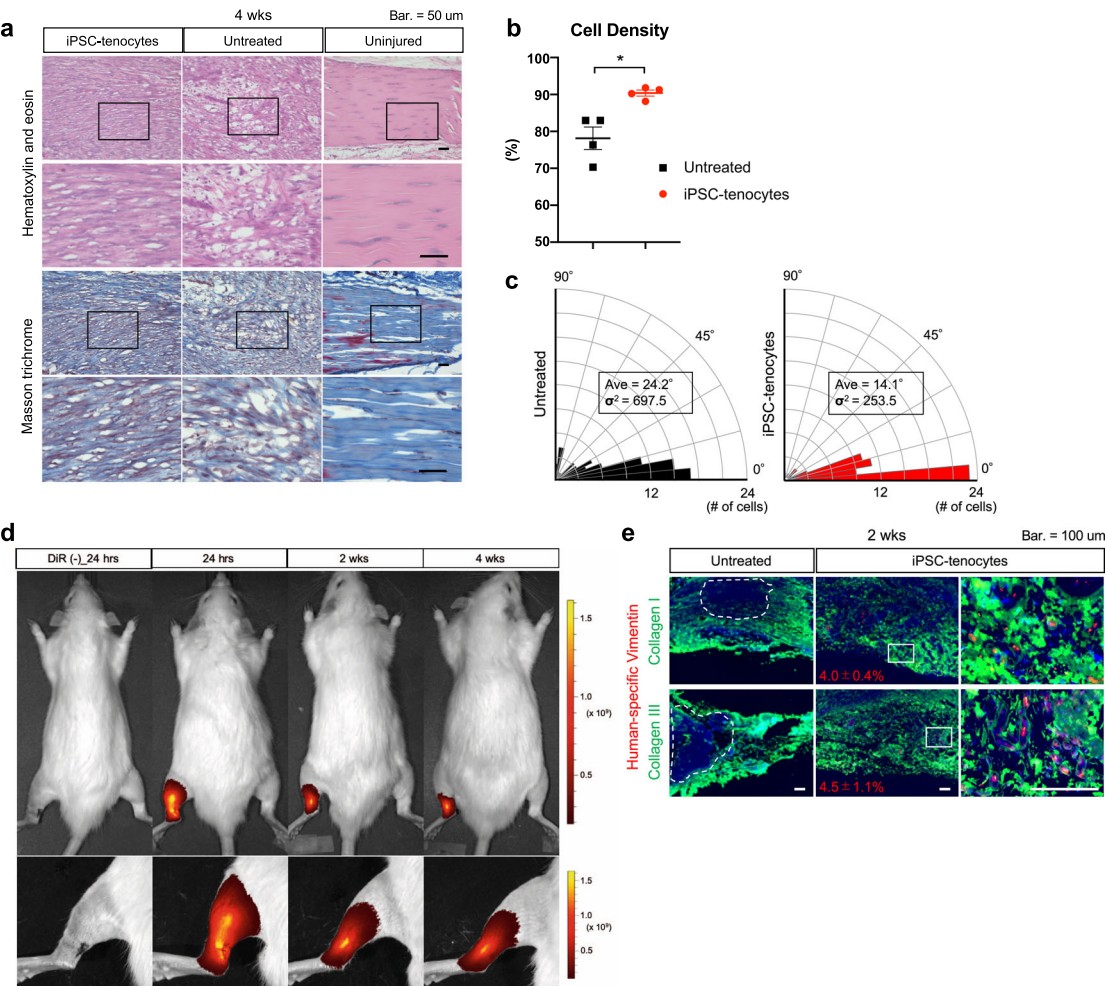

**Fig. 5 Engrafted iPSC-tenocytes express tendon functional extracellular matrix at two weeks after transplantation. a** Histological analyses of iPSC-tenocytes rats, untreated rats, and uninjured rats at 4 weeks after transplantation. Masson's trichrome staining shows collagen fiber (blue), cytoplasm (red), and nuclei (purple). Representative pictures of the transplanted area (left Achilles tendon) are shown. The boxed areas in the first and third rows are shown with higher magnification in the second and fourth rows, respectively. Scale bars: 50 μm. **b** Density of cell fibers stained with eosin. Image J was used for calculation ($n = 4$: biologically independent samples). Data represent mean ± SE. *$P < 0.05$, two-tailed Welch's t-test. **c** Rose diagrams of the regenerated fiber angle from the horizontal line. The angle of hematoxylin-positive cells was analyzed using Image J, and the average and variance were calculated. **d** Biodistribution of DiR-labeled iPSC-tenocytes at 24 h and 2 and 4 weeks after transplantation. **e** Immunohistochemistry of regenerated Achilles tendons harvested from iPSC-tenocytes rats and untreated rats at 2 weeks after transplantation. Tissues were stained with anti-human specific vimentin (red), type I collagen (green), and type III collagen (green) antibodies and co-stained with DAPI (blue). Representative images are shown. Dotted lines indicate non-regenerative regions. Positive rates of anti-human specific vimentin to DAPI were calculated using Image J ($n = 4$: biologically independent samples). Data represent mean ± SE. Scale bars: 100 μm. Source data are provided as a Source Data file.

tenocytes function as part of host tendon fibers themselves. In untreated rats, non-regenerative domains where collagen proteins were hardly expressed remained, whereas in iPSC-tenocytes rats, these proteins were uniformly stained, suggesting that a major function of transplanted iPSC-tenocytes is to promote host tendon regeneration. However, the engraftment rate seemed to be relatively low as only 4.0–4.5% of cells expressed vimentin at two weeks post-transplantation (Fig. 5e). These observations led us to speculate the existence of additional mechanisms of tendon regeneration besides donor cell engraftment.

**iPSC-tenocytes secrete growth factors that promote tenocyte proliferation and motor function recovery.** We collected a culture supernatant of iPSC-tenocytes and concentrated it by ultrafiltration to investigate the secreted proteins compared with those in the supernatant of cultured BMSCs. As shown in Fig. 6a, concentrated supernatant from iPSC-tenocytes greatly promoted

the proliferation of primary tenocytes in vitro compared to the flow-through, suggesting paracrine effects of iPSC-tenocytes. As the flow-through showed a stronger effect than the base medium, it could have been further concentrated. Proteomics analysis revealed that iPSC-tenocytes and BMSCs had 1,818 secreted common proteins (Fig. 6b), and the top 100 differentially expressed proteins between iPSC-tenocytes and sclerotome showed similar expression trends in BMSCs (Fig. 6c). iPSC-tenocytes expressed representative extracellular proteins and structural proteins of tendons, including collagens, SPARC, fibulins (FBN, FBLN), biglycan (BGN), and periostin (POSTN) (Fig. 6d). More interestingly, insulin-like growth factor (IGF) and TGFβ-related factors, such as IGF-binding proteins (IGFBP3, IGFBPL1, IGFBP2) and latent-transforming growth factor beta-binding protein-4 (LTBP4) were highly secreted by iPSC-tenocytes. As IGF and TGFβ are reportedly involved in tenocyte development and growth[41], we firstly evaluated their effects on primary tenocyte proliferation in vitro. Primary tenocytes

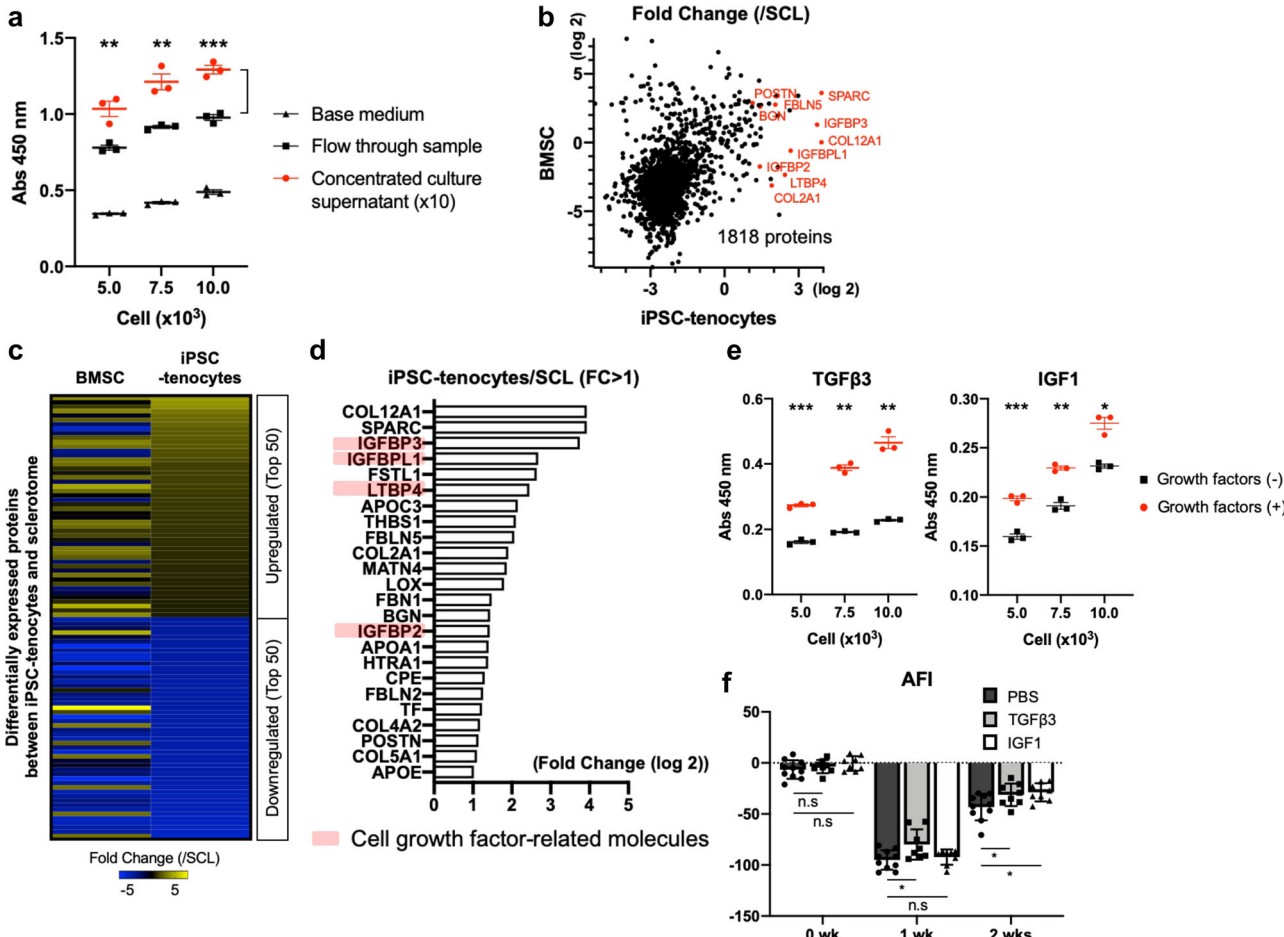

**Fig. 6 iPSC-tenocytes secrete cytokines that promote tenocyte proliferation and motor function recovery. a** Effect of concentrated culture supernatant of iPSC-tenocytes on primary tenocyte proliferation as assessed by cell proliferation assay. The statistical significance of the difference between the concentrated supernatant sample and the flow-through sample was assessed ($n = 3$: biologically independent samples). **b** Scatterplot distribution of 1,818 proteins detected in the culture medium of both iPSC-tenocytes and BMSCs. Fold changes (log 2) compared to iPSC-derived sclerotome are shown. **c** Heatmap showing the top 100 differentially expressed proteins in iPSC-tenocytes vs. iPSC-derived sclerotome. Protein expression in BMSCs is also shown for comparison. **d** List of extracellularly secreted proteins highly (log2[fold change] > 1) expressed in iPSC-tenocytes compared to iPSC-derived sclerotome. **e** Effects of exogenous TGFβ3 and IGF1 administration on tenocyte proliferation in vitro as assessed by cell proliferation assay ($n = 3$: biologically independent samples). **f** Kinematic analyses of rats treated with recombinant TGFβ3 or IGF1 proteins. AFI was measured at weeks 0 (before surgery), 1, 2 after transplantation (PBS: $n = 10$: biologically independent samples; TGFβ3 and IGF1: $n = 8$: biologically independent samples). *$P < 0.05$; **$P < 0.01$; ***$P < 0.001$ by multiple-comparisons two-tailed t-test (**a**) or two-tailed Welch's t-test (**e**, **f**). Data represent mean ± SE. n.s., not significant. Source data are provided as a Source Data file.

significantly proliferated in response to exogenous IGF1 and TGFβ3 (Fig. 6e). We then assessed the therapeutic effects of both factors by injecting respective recombinant proteins in rats after Achilles tendon rupture. Consequently, the AFI score of IGF1- or TGFβ3-treated rats was significantly restored at two weeks post-surgery (Fig. 6f), whereas no significant difference in ultimate failure load was observed at two weeks (Supplementary Fig. 4e). Analysis of strain-stress curves of the regenerated tendons showed that the breaking stress of TGFβ3-treated tendons tended to recover, although linear portion of the j-shaped curve did not show significant difference between the groups (Supplementary Fig. 4f–h). Overall, these findings imply that both IGF and TGFβ promoted host tenocyte proliferation in rats, contributing to motor function recovery after Achilles tendon rupture.

## Discussion

Cell therapy can be used for repairing damaged tissues. Two conceivable mechanisms may underlie the therapeutic effects: (1) engrafted donor cells complement tissue function by replacing damaged cells, (2) a paracrine effect exerted by donor cell-secreted growth factors promotes the host tissue's self-regenerative capacity. In this study, transplanted iPSC-tenocytes survived and expressed tendon-type collagens in host tissues at two weeks post-transplantation. Furthermore, proteomics analysis revealed that iPSC-tenocytes highly secreted representative extracellular and structural tendon proteins, which can reinforce the characteristics of the iPSC-tenocytes. More interestingly, iPSC-tenocytes secreted IGF- and TGFβ-related factors, such as IGFBP[42] and LTBP[43]. In line with the finding that IGF and TGFβ contribute to tenocyte development and proliferation[41], our results demonstrated that primary tenocytes significantly proliferated in response to exogenous IGF1 and TGFβ3 administration in vitro. Moreover, injecting the IGF1 or TGFβ3 recombinant proteins after Achilles tendon rupture contributed to motor function recovery in rats. The breaking stress of TGFβ3-treated tendons indicated a recovery trend at two weeks, whereas the ultimate failure load and linear portion of the j-shaped curve did not show a significant difference after IGF1 or TGFβ3

administration. Presumably, exogenous IGF1 reduced the functional deficit and accelerated motor function recovery via an anti-inflammatory mechanism, as previously described[44]. These observations suggest that the grafted iPSC-tenocytes contributed to regeneration in a dual manner: via engraftment and paracrine effects.

Early motor function recovery from a tendon injury is among the most valuable outcomes of this study, as it has not been reported in the field of PSCs[20]. It has great significance not only for athletes, but also for elderly people because prolonged rehabilitation often leads to bed-riddenness, which can cause disuse syndromes, such as dysphagia and dementia[45]. We believe that iPSC-tenocytes are a more promising option for cell therapy than adult primary cells or MSCs for two reasons. First, a large number of donor cells can be relatively easily prepared as iPSCs theoretically have unlimited proliferation capacity. In contrast, collecting primary cells or MSCs, especially from elderly people, often involves technical challenges. Second, cells derived from PSCs tend to have juvenile characteristics[46]. In cases where deterioration of tissue flexibility due to aging causes functional deterioration, grafting of juvenile cells would be a better treatment option[47]. In addition, we believe that this study contributes to the development of cell therapy because the human iPSC-tenocytes were differentiated in xeno-free conditions without any animal components. We highlight that the major benefits of our approach compared to those reported in previous studies on stem cell-derived tenocytes are the robustness and simplicity of our differentiation method.

We introduced an approach that models the normal progression of step-wise narrowing fate decisions during paraxial mesoderm development while inducing iPSCs toward tenocytes. As demonstrated by scRNA-seq, we derived PSM followed by somites, sclerotome, and eventually, differentiated tenocytes via syndetome. We believe this developmental biology-informed approach could contribute to the robust and efficient induction of tenocytes, as on day 19, a staggering 91.6% of cultured cells were positive for SCX, a pivotal gene for tenocyte development[48], and homogeneous, as evidenced by scRNA-seq. During the process of iPSC-tenocyte differentiation, while the gene expression of COL1A1 was low at PSM, SM, and SCL, COL1A2 started to express at the PSM stage (Supplementary Fig. 2a). These data suggest that the two alpha chains of type I collagen genes are differentially regulated during the early stage of iPSC-derived paraxial mesoderm differentiation. This finding aligns with several former studies demonstrating that certain cytokines, such as TGFβ1 and interleukin-4 (IL-4), can differentially regulate the transcription of each type I collagen gene in various cell types[49,50]. We believe that our method can also be a tool for studying genetic disorders related to tenocyte or ligament development, such as fibrodysplasia ossificans progressiva[51] and ossification of the posterior longitudinal ligament[52]. Utilizing patient-derived iPSCs, it would be possible to understand the pathology of disorders and screen for novel effective drugs.

However, this study had several limitations. Although histological analysis showed that tendon fibers treated with iPSC-tenocytes appeared well regenerated in terms of cell density and alignment, they were thinner than fibers in uninjured healthy rats, even at four weeks, probably suggesting inadequate regeneration. Second, considering the degeneration or shape-change of the Achilles tendon after injury, a better way to characterize the material properties of the regenerated tendon after iPSC-tenocyte transplantation would be to generate stress vs. strain curves and compare the slopes at the same level of strain (or stress). Third, it would have been valuable to check therapeutic effects on other tendons. For example, treating a severe injury of the rotator cuff by autologous transplantation is relatively difficult due to the complex structure and limited access; thus, cell therapy would offer a viable solution. We also envisage that cell therapy can be combined with surgical procedures to treat severe ruptures in the bone-tendon attachment sites. Fourth, to evaluate and control an inflammatory reaction after cell transplantation would reveal a possibility for enhanced therapeutic effects of iPSC-tenocytes since the inflammatory response plays a vital role in tendon healing[53]. Further investigations, such as grafting of rat PSC-derived tenocytes and combined use of immunosuppressive agents, would shed more light on the better therapeutic effects of iPSC-tenocytes transplantation. Finally, treatment of aged rats with iPSC-tenocytes, aiming for application in elderly people, remains to be assessed.

In summary, we utilized human iPSCs to recapitulate the embryonal development of the paraxial mesoderm and generated xeno-free tenocytes that showed a transcriptional profile resembling that of primary human tenocytes. We demonstrated that cell grafting contributed to efficient motor function recovery after Achilles tendon injury in rats via engraftment and paracrine effects. We suggest that human iPSC-tenocytes will provide a therapeutic approach for Achilles tendon injury.

## Methods

**Differentiation of tenocytes from iPSCs.** The human iPS cell line 1231A3[35] was established at Center for iPS Cell Research and Application, Kyoto University. The iPS cells were maintained under feeder-free and xeno-free. Briefly, the cells were cultivated on iMartix-511 (Nippi, Tokyo, Japan) ($0.5\,\mu g/cm^2$)-coated dishes with AK03 medium (Ajinomoto, Tokyo, Japan) and passaged via dissociation into single cells using Accutase (Thermo Fisher Scientific, Waltham, MA, USA). Three days prior to tenocyte induction, iPSCs were seeded onto iMartix-511-coated dishes ($1.1 \times 10^3$ cells/cm²) and cultured in AK03 medium. After three days, iPSCs were differentiated into the PSM in AK03 medium supplemented with 10 μM SB431542 (Sigma, St. Louis, MO, USA), 10 μM CHIR99021 (Wako, Osaka, Japan), 2 μM DMH1 (Tocris, Bristol, UK), and 20 ng/mL FGF2 (Wako) for four days[13,14]. For somite differentiation, four-day presomitic mesodermal cells were reseeded onto iMartix-511-coated dishes ($1.8 \times 10^4$ cells/cm²) and cultured in AK03 medium supplemented with 10 μM SB431542 and 5 μM CHIR99021 for four days[13,14]. On day 8, somites were cultured in AK03 containing 100 nM SAG (Calbiochem, La Jolla, CA, USA) and 0.6 μM LDN193189 (Stemgent, Cambridge, MA, USA) for sclerotome differentiation[13,14,36]. Eleven-day sclerotome cells were detached from the dish using Accutase and seeded onto iMatrix-511-coated dishes ($1.8 \times 10^4$ cells/cm²), and subsequently differentiated toward syndetome in AK03 medium supplemented with 10 ng/mL FGF8 (Peprotech, Rocky Hill, NJ, US) and 10 ng/mL TGFβ3 (R&D Systems, Minneapolis, MN, USA) for the first two days, and in AK03 medium supplemented with 10 ng/mL TGFβ3 and 10 ng/mL BMP7 (R&D Systems) for the next six days.

**Human primary tenocytes and MSCs.** Human primary tenocytes and human MSCs from BMSCs were purchased from Angio-Proteomie (Boston, MA, USA) and PromoCell (Heidelberg, Germany) and cultured in Tendon Cell Growth Medium (Angio Proteomie) and Mesenchymal Stem Cell Growth Medium (PromoCell), respectively.

**RT-qPCR analysis.** Total RNA was isolated using a RNeasy Kit (Qiagen, Hilden, Germany), and genomic DNA was removed with a DNase-one Kit (Qiagen). RT was carried out using 1 μg of total RNA and Superscript III reverse transcriptase (Invitrogen, Carlsbad, CA, USA), according to the manufacturer's instructions. qPCRs were run using Thunderbird SYBR qPCR Mix (Toyobo, Osaka, Japan) in a QuantStudio12K Flex Real-Time PCR system (Applied Biosystems, Foster City, CA, USA) or a StepOne real-time PCR system (Applied Biosystems). Primer sequences are listed in Supplementary Table S1.

**Immunocytochemistry, immunohistochemistry, histological analyses, and EdU staining.** For immunocytochemistry, cultured cells were fixed with 2% paraformaldehyde at 4 °C for 10 min, washed twice with PBS, and incubated with 0.2% Tween-20 (Sigma) in PBS at 4 °C for 15 min for penetration processing. Subsequently, the cells were treated with Blocking One (Nacalai Tesque, Kyoto, Japan) at 4 °C for 1 h and incubated with primary antibodies at 4 °C overnight. Next, the cells were washed several times with 0.2% Tween-20/PBS and incubated with secondary antibodies at room temperature for 1 h. DAPI (1:5000; Sigma) was used to counterstain the nuclei. The primary and secondary antibodies are listed in Supplementary Table S2. Immunohistochemistry and histological analysis of animal tissues were carried out at the Center for Anatomical, Pathological, and Forensic Medical Research, Graduate School of Medicine, Kyoto University. The stained

samples were observed and analyzed under a fluorescence microscope (BZ-X700; Keyence, Osaka, Japan). The density and alignment of regenerated tendon fibers were analyzed using Image J software. EdU staining was carried out with an EdU staining proliferation kit (ab219801, Abcam, Cambridge, MA, USA) according to the manufacturer's instruction. Briefly, iPSC-tenocytes were fixed and then permeabilized before staining. A reaction mixture containing fluorescently labeled EdU was added, and the cells were incubated for 30 min.

**Single-cell RNA sequencing**. Dissociated human iPS cells or differentiated cells were resuspended with 1 mL of 0.04% BSA in PBS. Each single-cell suspension containing approximately $2.0 \times 10^3$ cells with enzyme mix, gel beads, and oils were loaded onto 10× Genomics chromium chips to generate Gel Bead in Emulsion. Subsequently, RT and cDNA amplification were conducted according to the manufacturer's instruction for single-cell 3′ reagent kits v3 (10x Genomics, Pleasanton, CA, USA). The cDNA libraries were sequenced on the HiSeq 2500 platform (Illumina, San Diego, CA, USA). The sequence reads derived from each sample were validated by each barcode and aligned to the human genome reference GRCh38 using CellRanger software v3.1.0. The raw data of count matrices showing gene expression in the cells obtained from each culture condition calculated by CellRanger were combined using Seurat v3.1.1 R packages on R v3.6.1. Unique molecular indices indicating <250 genes, <10,000 reads, or >25% of mitochondrial transcripts were filtered out. Subsequently, the combined count data were normalized and subjected to principal component (PC) analysis. Twenty PCs were used for UMAP to visualize the relationship among the cells and clustering analysis. The cell cycle scores of each cell were assigned by referring to the cell cycle-related gene list in a previous report[54]. Data mining for cell-type-specific genes and differentially expressed genes among two groups were conducted according to Seurat vignettes (https://satijalab.org/seurat/vignettes.html).

**iPSC-tenocyte transplantation, and IGF1 and TGFβ3 injection after Achilles tendon rupture**. Eight-week-old male F344/Nslc rats were purchased from Shimizu Laboratory Supplies (Kyoto, Japan) and acclimatized for several days. The rats were randomly divided into eight treatment groups: iPSC-tenocytes group, iPSC-derived sclerotome group, BMSCs group, untreated group (incised, but not transplanted), uninjured group (neither incised nor transplanted), IGF1 injection group, TGFβ3 injection group, and PBS injection group. Rats weighing 130–170 g were anesthetized with 3% isoflurane (Pfizer Japan, Tokyo, Japan) in an anesthetic chamber and then received an intraperitoneal injection of 0.375 mg/kg medetomidine hydrochloride (Nippon Zenyaku Kogyo, Fukushima, Japan), 2 mg/kg midazolam (Sandoz, Tokyo, Japan), and 2.5 mg/kg butorphanol tartrate (Meiji Seika Pharma, Tokyo, Japan) mixed anesthetic. To injure the left Achilles tendons, a complete transverse incision at 5 mm from the calcaneus over the Achilles tendon was made with a surgical blade, and the skin was sutured using 5-0 monofilament nylon. Immediately after the operation, 50 μL of DMEM/F12 (Gibco, Grand Island, NY, USA) diluted with Matrigel (BD Biosciences, Franklin Lakes, NJ, USA) (v/v = 1:1) containing $3 \times 10^6$ iPSC-tenocytes, sclerotome, or BMSCs was injected into the injured site using a 0.5-mL syringe. In a control experiment, 50 μL of DMEM/F12 diluted with Matrigel (v/v = 1:1) without cells was injected using the same method. In Fig. 6f and Supplementary Fig. 4e-h, 10 μg IGF1 diluted in 50 μL PBS/Matrigel (1:1) or 10 ng TGFβ3 in 50 μL PBS/Matrigel (1:1), or 50 μL PBS/Matrigel (1:1) was injected using the same method on day 0: soon after the operation, on days 1, 2, 8, and 12. Shortly after the operation, the rats were placed inside a weight-bearing orthosis for tail suspension for 1 week, and tendon regeneration was assessed by kinematic testing, biomechanical testing, and histological testing at 1, 2, 3, and 4 weeks after surgery.

**AFI calculation**. A paper tape was placed at the bottom of a walking alley, which was bilaterally walled by plastic plates, with a darkened box at the end, as reported[55]. Ink was smeared over the planta pedis of a rat using a brush, and then the animal was placed at the entrance of the alley. The rat left footprints as it walked straight through the alley into the darkened box. For each rat, three pairs of legible footprints were randomly selected, and three parameters were measured: print length (PL): length from the heel end to the tip of the third toe; toe spread (TS): distance from the first toe to the fifth toe; intermediate toe spread (IT): distance from the second toe to the fourth toe. The AFI was calculated using the following formula (1)[40]:

$$\text{AFI} = 74[(\text{NPL} - \text{EPL})/\text{EPL}] + 161[(\text{ETS} - \text{NTS})/\text{NTS}] + 48[(\text{EIT} - \text{NIT})/\text{NIT}] - 5 \tag{1}$$

where E is the experimental side (left leg), and N is the normal side (uninjured right leg). A value of 0 indicates the normal function, and a negative value indicates impairment.

**Kinematic testing**. As previously described[55], rats were anesthetized with 3% isoflurane in an anesthetic chamber, and different-colored hemispheric markers were bilaterally attached to three landmarks on the shaved skin, including the knee joint, calcaneus, and fifth metatarsal. Subsequently, the rats were placed on a treadmill (12 m/min) with a 3-dimensional (3D) motion capture apparatus (KinemaTracer; Kissei Comtec, Nagano, Japan) and at least 10 consecutive steps

were recorded for each rat, using a camera. Using the built-in software (Kissei Comtec), 3D kinematic models based on tracing markers were built to measure two parameters, i.e., heel height on foot contact: vertical distance from the fifth metatarsal to the calcaneus, and ankle angle on foot contact: the angle formed by a line connecting the knee joint to the calcaneus and by a line connecting the calcaneus to the fifth metatarsal. A positive angle indicates dorsal flexion, and a negative angle indicates plantar flexion.

**Biomechanical testing**. Regenerated Achilles tendons, which attach the gastrocnemius muscle to the calcaneus, were harvested at 2 and 4 weeks. Muscle segments were frozen in liquid nitrogen. Muscle-tendon-bone units were fastened in a clamping device and mounted onto a mechanical testing machine (Autograph AG-X; Shimadzu, Kyoto, Japan). The units were pulled from both sides at a constant speed (200 mm/min) until failure, and the ultimate failure load (N) was measured. All regenerated tendons used in the test broke in the midsubstance.

**In vivo imaging**. iPSC-tenocytes were incubated with 320 μg/mL of XenoLight 1,1′-dioctadecyltetramethylindotricarbocyanine iodide (DiR; PerkinElmer, Hopkinton, MA, USA) for 30 min to evaluate the engraftment of the donor cells. To visualize DiR fluorescence in vivo, rats were anesthetized in a chamber with 3.0% isoflurane after shaving to avoid light scattering and were then placed in an IVIS SpectrumCT In Vivo Imaging System (PerkinElmer). The biodistribution of DiR-labeled iPSC-tenocytes was monitored at 24 h, 2 weeks, and 4 weeks after transplantation, and the captured data were analyzed with Living Image Software 4.5 (PerkinElmer).

**Ultrafiltration of iPSC-tenocyte supernatant**. Culture supernatants of iPSC-tenocytes were collected and ultrafiltrated using centrifugal filters (Merck Millipore, Burlington, MA, USA) to concentrate supernatant proteins with a membrane nominal molecular weight limit of 10 kDa. The concentrated proteins and the flow-through were collected. Protein concentration was assessed by the BCA assay using Pierce BCA Protein assay Reagents (Thermo Fisher Scientific).

**Sample preparation for MS analysis**. Culture supernatants of iPSC-tenocytes and BMSCs were replaced with PBS three days before sample preparation. An equal volume of lysis buffer (24 mM SDC, 24 mM SLS, 200 mM Tris-HCl (pH 9.0), 2% phosphatase inhibitor, and 2% protease inhibitor) was added to the collected PBS. After treating the samples at 95 °C for 5 min, the samples were subjected to reduction, alkylation, Lys-C/trypsin digestion (enzyme ratio: 1/100), and desalting[56]. The resulting peptides were labeled with isobaric tags for relative and absolute quantification (iTRAQ, Sciex, Framingham, MA, USA). Briefly, 120 μg of desalted peptide samples were dried and dissolved in 10 μL of 500 mM triethylammonium bicarbonate. Approximately 20 μL of iTRAQ reagents (Multiplex kit, Sciex) was added to 23 μL of ethanol and mixed with the peptide sample. After incubation at room temperature for 1.5 h, 16 μL of 10% TFA and 400 μL of loading buffer (0.5% trifluoroacetic acid and 4% (v/v) acetonitrile) were added to quench the reaction, and the sample mixture was desalted using StageTip[57]. Then, the samples (8 μg) were subjected to nanoLC-MS/MS analysis.

**NanoLC-MS/MS analysis**. NanoLC–MS/MS was carried out using a TripleTOF 5600 System (AB, Sciex) equipped with an HTC-PAL autosampler (CTC Analytics). The peptides were separated on a monolithic column (4 m, 100 μm i.d., GL Science, Tokyo, Japan) using a Dionex UltiMate 3000 RSLCnano System. The mobile phases were 0.5% acetic acid with 5% (v/v) DMSO (solution A) and 0.5% acetic acid in 80% (v/v) acetonitrile with 5% (v/v) DMSO (solution B)[58]. For elution, 5–15% solution B for 205 min, 15–35% solution B for 549 min, 35–40% solution B for 103 min, 40–100% solution B for 5 min, 100% solution B for 118 min and 5% solution B for 100 min was used (total, 1,080 min), with a flow rate of 400 nL/min. The coiled monolithic capillary column was connected to a self-pulled emitter (100 μm i.d., 3–5-μm tip) formed with SutterP-2000 and a conductive distal coating end applied with an ion coater model IB-2 (Eiko Engineering, Ibaraki, Japan) from which the spray voltage was applied. The applied spray voltage was 2,300 V, and the interface heater temperature was 150 °C. The MS scan range was 300–1,500 m/z every 0.25 s, and the MS/MS scan range was 80–1500 m/z every 0.1 s. The maximum number of candidate ions monitored per cycle was 10, and the cycle time was 1.3 s. The resolution of the Q1 scan was UNIT. To minimize repeated scanning, previously scanned ions were excluded for 30 s. Analyses were performed in duplicate, and blank runs were inserted between samples.

**Proteome data analysis for protein identification**. The raw data files were analyzed using ProteinPilot v5.0 (Sciex) with acceptable modifications of N-terminal iTRAQ, iTRAQ of lysine, carbamidomethylation of cysteine, oxidation of methionine, phosphorylation of serine, threonine, or tyrosine, deamidation of asparagine or glutamine, N-terminal pyro-glutamic acid of glutamine or glutamic acid, and protein N-terminal acetylation. Peak lists generated from a ProteinPilot.group file were analyzed using Mascot v2.5 (Matrix Science, London, UK), with carbamidomethylation of cysteine as fixed modification, and N-terminal iTRAQ, iTRAQ of lysine, and methionine oxidation as variable modifications. Database

searches against selected human entries of UniProt/Swiss-Prot release 2016_06 (8-June-2016) were conducted, with a precursor mass tolerance of 20 ppm, a fragment ion mass tolerance of 0.1 Da, and strict trypsin and Lys-C specificity, which allowed up to two missed cleavages. For peptide identification, peptides were rejected if any of the following conditions were not satisfied: (a) if the same scan was assigned to different peptides between ProteinPilot and Mascot, (b) peptide confidence was < 0.05, (c) charge state was >5, and (d) peptide length was <6 amino acids. At least two confidently ($P < 0.05$) identified peptides per protein were used for protein identification. Single peptides with higher confidence ($P < 0.01$) were allowed. Finally, peptides were grouped into protein groups based on previously established rules[59]. False discovery rates were estimated by searching against a decoy sequence database (<1%). For peptide and protein quantification, the iTRAQ area was normalized to the total area of the whole proteome for each sample, and we used the RiMS approach to increase accuracy[60].

**Cell proliferation assay**. Human primary tenocytes ($5.0 \times 10^3$, $7.5 \times 10^3$, and $10.0 \times 10^3$) were seeded into a 96-well plate containing DMEM/F12 (Gibco), and after 24 h, 10 ng/mL TGFβ3 or 50 ng/ml IGF1 was added. Cell proliferation was assessed after 72 h using Cell Counting Kit-8 (Dojindo Molecular Technologies, Kumamoto, Japan) according to the manufacturer's instructions.

**Statistics and reproducibility**. Statistical analyses were conducted using Graph-Pad Prism 8 (GraphPad Software, La Jolla, CA, USA). $P < 0.05$ was considered significant. Representative pictures are shown in immunocytochemistry, immunohistochemistry, and histological analyses such as Figs. 1d, 5a, Supplementary Fig 4c, d. These experiments were repeated independently at least three times and confirmed similar results.

**Study approval**. All animal experiments were approved by the Institutional Animal Committee of Kyoto University (No. 16-73-11).

**Reporting summary**. Further information on research design is available in the Nature Research Reporting Summary linked to this article.

## Data availability

The accession number for single-cell RNA sequencing data reported in this study is "GSE156753". The MS/MS data were deposited in ProteomeXchange Consortium via jPOSTrepo[61] with a dataset identifier "JPST000939". All other relevant data supporting the key findings of this study are available within the article and its Supplementary Information files or from the corresponding author upon reasonable request. Source data are provided with this paper.

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

## Acknowledgements

We thank Drs. D. Murata, D. Kamiya, Y. Toyooka, and N. Boyd-Gibbins for invaluable discussions, comments, and critical reading of the manuscript, M. Terashima, and Y. Inada for technical support, Dr. M. Nomura, K. Ohnishi, and J. Kuwahara for single-cell RNA sequencing, and all members of the Ikeya laboratory for their support in this study. This work was supported by the Japan Society for the Promotion of Science (JSPS) KAKENHI grant number 20H03803 (to M.Ik.), the Core Center for iPS Cell Research of the Research Center Network for Realization of Regenerative Medicine (20bm0104001h0008) and the Projects for Technical Development, which is a program of the Research Center Network for Realization of Regenerative Medicine (20bm0404066h0001), from the Japan Agency for Medical Research and Development (AMED) (to M.Ik.), a grant from the iPS Cell Research Fund (to M.Ik.), and a research grant from the Fujiwara Memorial Foundation (to T.N.).

## Author contributions

T.N.: Conception and design, data collection and assembly, data analysis and interpretation, manuscript writing, final approval of the manuscript; A.N., N.Y., K.Y., C.Z.: Data collection and assembly; T.K.: Analysis and interpretation of single-cell RNA sequencing data; M.Iw.: experimental design and proteome data interpretation; H.K.: administrative support; M.Ik.: Conception and design, financial support, administrative support, data interpretation, manuscript writing, final approval of the manuscript.

## Competing interests

The authors declare no competing interests.
