## [Peer Review File · Nature Communications]

Reviewers' Comments:

Reviewer #1:

Remarks to the Author:

The manuscript by Nakajima et al describes the use of tenocytes derived from human iPSCs to aid in the functional recovery of rats with Achilles tendon ruptures. There is definitely a need to develop better therapies to repair damaged tendons and using iPSCs to generate tenocytes for implantation, rather than endogenous tendon cells that are not in high abundance, is an approach worth investigating further. The authors are using a slight modification of their previously reported protocol that uses developmental cues to drive human iPSCs towards different derivatives of the somites. The iPSC tenocytes are then used to promote recovery in a rat tendon model. This reviewer appreciates the study tested for functional recovery of the rats' ambulatory abilities rather than just restoration of the mechanical properties of the tendon tissue. The conclusion that the effect of the implanted cells may be due to paracrine factor secretion due to the low number seems valid and the approach to identify what those factors may be by performing proteomics analyses was nice. However, additional confirmation is needed to determine the effect of iPSC-tenocytes on Achilles tendon healing in vivo; simply showing that IGF and TGFb3 promotes cell proliferation in vitro is not convincing that it is a paracrine effect. Due to this concern as well as those delineated below, it is recommended that the manuscript undergo a major revision before reconsideration for publication.

1. From what this reviewer can tell, the most of the results in Figure 1 were previously reported by the authors in Nakajima et al., Development 2018, with the only difference being that the cells in this manuscript are cultured in xeno-free conditions. The difference between these two protocols needs to be better described, or this information should be relegated to the supplemental info.

2. Perhaps this is due to this reviewer's naivete, but it is unclear why there are two different UMAP plots in Figure 3a. In the manuscript, the [0,1,2,3] + [4,7] clusters are defined as the syndetome. Why not just refer to them as two different clusters? What is the physiological difference between the cell populations in clusters 1 and 2? This is not discussed. In addition, the color schemes should be made distinct so that the reader does not attempt to correlate colors between the two graphs (e.g. pink is for primary human tenocytes in the left plot, but then only cluster 10 is pink in the right plot).

3. It is stated "These data demonstrated that the iPSC-tenocytes were a homogeneous population that covered various cell cycle phases, and their transcriptional features showed some similarities to those of primary tenocytes," (P7, lines 160-162); however, there is no data demonstrating this. A comparison of the top 20 similarly and differentially expressed genes between [0,1,2,3] + [4,7] and [8, 10] clusters, similar to what was done in 3d, S1b needs to (and should) be included to make any conclusion about the similarities of the iPSC-tenocytes to the primary tenocytes.

4. Reporting of the load to failure (Figure 4d) is not informative about whether or not the tendon material has been restored. It is recommended that the stress-strain curves be reported so that it can be determined that repaired tendon has the same properties as the uninjured control. Fortunately, the functional assays (Figure 4c, d(left, middle)) indicate that is the case. But the reason that the non-treated was able to withstand the same amount of force is likely because the damaged area was not replaced with functional tendon/ECM material, just a lot of scar tissue (and perhaps the reason the functional assays show reduced recovery for the non-treated).

5. The functional recovery from iPSC-tenocytes should be plotted along with the iPSC-sclerotome and BMSC transplantation in Figure S2 to help the reader directly compare this data and make their own conclusions. In addition, the percent of tendons treated with iPSC-sclerotome cells that had excess glycosaminoglycan staining needs to be stated. It cannot be concluded that increase in alcian blue means that there is cartilage tissue unless cartilage-specific markers are included. This needs to be restated.

6. Additional confirmation of the paracrine role of the iPSC-tenocytes needs to be conducted beyond adding TGFb3 and IGF to induce proliferation of tenocytes in vitro (both have already been

demonstrated for various types of fibroblasts in the literature). For example (as suggested in the limitations), direct assessment of IGF and TGFβ3 in vivo by injecting the growth factors or manipulating the iPSC-tenocytes so these factors are over or under expressed will support the conclusions. Another way is to assess for the increased expression of human IGF or TGFβ3 in these tendons in vivo compared with untreated. Without additional confirmation, the conclusion that paracrine factors from the iPSC-tenocytes are affecting the healing is highly correlative and weak.

7. The writing can be made clearer. There are abrupt transitions between some sentences/ideas. For example, the first three sentences of the introduction are awkward and can use better transitions.

8. COL1A2 might be mislabeled as COL2A1 in Figure 2d, or alternatively COL2A1 is mislabeled as COL1A2 in Figure S1a. It is unclear why COL1A2 would be highly expressed at PSM/somite/SCL stages when COL1A1 is not considering these two protein chains complex into a heterotrimer to form type I collagen fibrils. These should be double checked and if there is a discrepancy in when the two alpha chains of type I collagen are differentially regulated at during syndetome differentiation, it should be briefly discussed.

Reviewer #2:

Remarks to the Author:

The manuscript (#278068) of Nakajima et al. is yet-another tour de force study from this group to manipulate embryonic signaling pathways in sequential combinations and generate tenocytes from human iPSCs. Using sc-RNA-seq across the time line of tenocyte-derivation, they map out the developmental trajectory, which provides an informative foundation for in vivo investigation. They further show that iPSC-derived tenocytes can contribute to tendon proper when transplanted into injured Achilles tendon in rats. All together this is a strong study with critical information that will benefit the tendon development and repair fields for the time to come. With that said, I have 2 broad comments.

1. Studies in mouse and zebrafish have shown that embryonic and perinatal (larval and even adult fish) tenocytes possess progenitor properties in that they are able to proliferate. Hence, the authors rightfully aimed for the embryonic tenocytes with the goal to help repair tendon. It will be nice to exploit that property in this culture system. For example, what is the proliferative limitation of these in vitro iPSC-derived tenocytes/progenitors over time? Are there a mixture of progenitors and 'terminally differentiated tenocytes'? What are their relative proportions? While the scRNA-seq data give hints, a direct demonstration by cell biological data will be very helpful (e.g. using the markers they assigned for the stages/states of progression coupled with BrdU). These results may also help to explain the limited engraftment.

2. While this study represents a leap forward for tenocyte development, the goal for tendon healing is a bit disappointing: At 4 weeks, there is no improvement over untreated (please change 'non-treat' to untreated, and 'non-injure' to uninjured) by a variety of measurements (Fig. 4), albeit a faster recovery at 2 weeks. Given the cell density increase and poor engraftment, one wonders where do the new cells come from? It was proposed, based on the secretome, that paracrine signaling by TGFβ3 and IGF1 is responsible. This does makes sense based on previous studies. But, why not take this a step further by applying these two factors to injured tendons? I understand that the authors may wish to test this proposal in a subsequent study to elucidate their individual and combined roles in tendon healing extensively. However, as a proof of principle study, they should include the positive effect of these two factors, even without knowing their target cell types – e.g. it could be the existing tenocytes, stem cells, or an unknown/unexpected cell type.

Reviewer #3:

Remarks to the Author:

This is an excellent and innovative work that opens a new range of possibilities in the cell therapy/regenerative medicine field to treat tendon injuries more efficiently. The claims are very

well supported by the findings/conclusions of the study and the manuscript itself is very clearly and concisely written. It is also appreciated that the authors recognise and discuss very honestly the limitations of the study. The discussion of the results could improve significantly if the authors could add further discussion on the following points:

- While the protocol developed to obtain the iPSCs-tenocytes is undoubtedly robust, it is also complex, requiring several steps, relatively long times of culture and expensive reagents. How do the benefits of this approach compare to tenocytes derived/differentiated from stem cell sources that have been reported?
- The authors mentioned the relevance of addressing the proposed cellular therapy in other tendons, namely in the rotator cuff, a major clinical problem. How do the authors envisage the application of a cellular therapy alone to solve injuries that are so demanding in terms of their biomechanical requirements?

Could authors also explain if, given that it is well-known that the inflammatory response plays a key role in tendon healing/regeneration, the response observed in the rat model selected could be different if implanted with rat cells instead of human derived cells?

Reviewer 1**Comment 1:**

From what this reviewer can tell, the most of the results in Figure 1 were previously reported by the authors in Nakajima et al., Development 2018, with the only difference being that the cells in this manuscript are cultured in xeno-free conditions. The difference between these two protocols needs to be better described, or this information should be relegated to the supplemental info.

Answer to Comment 1:

A major difference from our previous methods is a timing of TGFb3 treatment as we administrated TGFb3 at day 3 of syndetome differentiation in the original methods. We newly described this point in the revised manuscript on line 109-111.

Comment 2:

Perhaps this is due to this reviewer's naivete, but it is unclear why there are two different UMAP plots in Figure 3a. In the manuscript, the [0,1,2,3] + [4,7] clusters are defined as the syndetome. Why not just refer to them as two different clusters? What is the physiological difference between the cell populations in clusters 1 and 2? This is not discussed. In addition, the color schemes should be made distinct so that the reader does not attempt to correlate colors between the two graphs (e.g. pink is for primary human tenocytes in the left plot, but then only cluster 10 is pink in the right plot).

Answer to Comment 2:

We added color schemes and titles for each UMAPs in Fig.3a to prevent misreading. Regarding the physiological difference in those syndetome clusters (e.g. cluster 1 and cluster 2), we're considering that the iPSCs-syndetome are homogenous populations as tenocytes-related gene markers including *SCX* and *COL1A2* were universally expressed throughout syndetome clusters (0,1,2,3,4,7) as shown in Fig.3b. As you suggested, we newly analyzed the top20 differentially expressed genes between cluster 1 and cluster 2 (as shown below). While some tenocyte-related ECMs such as collagens and *POSTN* were listed in top20 which possibly implies the slight differences in cell maturity between cluster 1 and cluster 2, we think that the difference is considered to be really low as the dot plot analysis showed. We would like to keep the original numbering of each cluster for the sake of convenience considering the other analysis such as Fig.3e.

Comment 3:

It is stated “These data demonstrated that the iPSC-tenocytes were a homogeneous population that covered various cell cycle phases, and their transcriptional features showed some similarities to those of primary tenocytes,” (P7, lines 160-162); however, there is no data demonstrating this. A comparison of the top 20 similarly and differentially expressed genes between [0,1,2,3] + [4,7] and [8, 10] clusters, similar to what was done in 3d, S1b needs to (and should) be included to make any conclusion about the similarities of the iPSC-tenocytes to the primary tenocytes.

Answer to Comment 3:

We agree with your proposal that to further investigate the similarities between iPSC-syndetome and primary tenocytes. Instead of making a list of top20 similarly expressed genes in syndetome and primary tenocytes, we analyzed top 40 differentially

expressed genes in human primary tenocytes vs. iPSC-sclerotome then assessed those gene expression patterning in syndetome (Fig. 3d). In this case, we think that putting sclerotome as a reference sample is necessary to characterize the similarities between syndetome and primary tenocytes. Hence, we firstly made a list of “top20 upregulated gene in primary tenocytes compared to sclerotome (Teno/SCL)” and “top20 downregulated gene in primary tenocytes compared to sclerotome (Teno/SCL)”, then assessed those 40 genes expression in syndetome. As you see in Fig. 3d, heatmap showing the top 40 differentially expressed genes in human primary tenocytes vs. iPSC-derived sclerotome exhibited similar expression trends in syndetome and primary tenocytes. The heatmap also indicates the gradual differentiation approach from sclerotome toward tenocytes as the color patterning shows. We described these data in the revised-manuscript on line 157-161.

Furthermore, as for tenocyte-specific extracellular matrix-related gene expression, syndetome exhibited expression characteristics closer to primary tenocytes than to sclerotome (Fig. 3c). With these data, we consider that the iPSC-syndetome and primary tenocytes share some transcriptional similarity.

Comment 4:

Reporting of the load to failure (Figure 4d) is not informative about whether or not the tendon material has been restored. It is recommended that the stress-strain curves be reported so that it can be determined that repaired tendon has the same properties as the uninjured control. Fortunately, the functional assays (Figure 4c, d(left, middle)) indicate that is the case. But the reason that the non-treated was able to withstand the same amount of force is likely because the damaged area was not replaced with functional tendon/ECM material, just a lot of scar tissue (and perhaps the reason the functional assays show reduced recovery for the non-treated).

Answer to Comment 4:

We appreciate your suggestion. To further assess the properties of repaired tendon, we newly analyzed the load-displacement curves that are generally used to judge the properties of substances. We created a new supplemental figure as Supplementary Fig. 3a and showed a representative load-displacement curve of each repaired tendon 2 weeks after transplantation: iPS-tenocytes group, untreated group, and uninjured group. Untreated rats showed a gentle curve that indicates high viscosity of the repaired tendon. Since high viscosity of tendon can adversely affect the muscle function, we think that functional assays showed a reduced recovery in the untreated group (Fig4c, 4d). On the other hand, iPS-tenocytes rats and uninjured rats showed similar curves with a sharp drop immediately after the rupture. These results imply that regenerated tendon in iPS-tenocytes group has

been restored in terms of physical characteristics. We described these data in the revised-manuscript on line 200-206 as well.

Comment 5:

The functional recovery from iPSC-tenocytes should be plotted along with the iPSC-sclerotome and BMSC transplantation in Figure S2 to help the reader directly compare this data and make their own conclusions. In addition, the percent of tendons treated with iPSC-sclerotome cells that had excess glycosaminoglycan staining needs to be stated. It cannot be concluded that increase in alcian blue means that there is cartilage tissue unless cartilage-specific markers are included. This needs to be restated.

Answer to Comment 5:

We added the datasets of iPSC-tenocytes in Supplementary Fig. 4a,b. Also, we included an explanation about the percent of alcian blue staining in the revised-manuscript on line 216-217. In addition, we performed immunohistochemistry and observed SOX9 expression in the regenerated Achilles tendons harvested from iPSC-derived sclerotome rats at 4 weeks after transplantation. Thus, we concluded that ectopic cartilage tissue developed in the host tendon fibers at 4 weeks after iPSC-derived sclerotome transplantation. This staining data was newly added as Supplementary Fig. 4d, and explained in the manuscript on line 213-216.

Comment 6:

Additional confirmation of the paracrine role of the iPSC-tenocytes needs to be conducted beyond adding TGFb3 and IGF to induce proliferation of tenocytes in vitro (both have already been demonstrated for various types of fibroblasts in the literature). For example (as suggested in the limitations), direct assessment of IGF and TGFb3 in vivo by injecting the growth factors or manipulating the iPSC-tenocytes so these factors are over or under expressed will support the conclusions. Another way is to assess for the increased expression of human IGF or TGFb3 in these tendons in vivo compared with untreated. Without additional confirmation, the conclusion that paracrine factors from the iPSC-tenocytes are affecting the healing is highly correlative and weak.

Answer to Comment 6:

We assessed the therapeutic effects of exogenous TGFb3 and IGF1 in vivo by injecting respective recombinant proteins after Achilles tendon rupture in rats. As shown in Fig. 6f, the AFI score of rats treated with IGF1 or TGFb3 was significantly restored at 2 weeks post-surgery compared to the control rats that were treated with PBS. Although the breaking stress of TGFb3 treated tendon indicated a recovery trend at 2 weeks, the ultimate

failure load did not show a significant difference after IGF1 nor TGFb3 administration (Supplementary Fig. 4e). We presume that exogenous IGF1 reduced functional deficit and accelerated motor function recovery via an anti-inflammatory mechanism as described in a previous work in an Achilles tendon injury rat model (C. Kurtz et al., The American Journal of Sports Medicine, 1999). With these data, we conclude that paracrine factors: IGF1 and TGFb3 contributed to the recovery of motor function after Achilles tendon rupture. We also assume that cell transplantation possesses a better therapeutic effect in comparison with the simple injection of IGF1 and TGFβ3 recombinant proteins. These data were newly added as Fig. 6f and Supplementary Fig. 4e, and explained in the manuscript on line 262-269, 283-290.

Comment 7:

The writing can be made clearer. There are abrupt transitions between some sentences/ideas. For example, the first three sentences of the introduction are awkward and can use better transitions.

Answer to Comment 7:

We made several changes in the manuscript to make writing clearer. We also requested a native speaker of English to proofread our English writing.

Comment 8:

COL1A2 might be mislabeled as COL2A1 in Figure 2d, or alternatively COL2A1 is mislabeled as COL1A2 in Figure S1a. It is unclear why COL1A2 would be highly expressed at PSM/somite/SCL stages when COL1A1 is not considering these two protein chains complex into a heterotrimer to form type I collagen fibrils. These should be double checked and if there is a discrepancy in when the two alpha chains of type I collagen are differentially regulated at during syndetome differentiation, it should be briefly discussed.

Answer to Comment 8:

We confirmed that there is no typo in Fig. 2d and Supplementary Fig. 2a. We think that the two alpha chains of type I collagen genes can be differentially regulated during the early stage of iPSC-derived paraxial mesoderm differentiation as several studies demonstrated that some cytokines such as TGFb1 and interleukin-4 (IL-4) can differentially regulate the transcription of each type I collagen gene in various cell types (Fertin C et al., Cell Mol Biol 37(8), 1991; Sempowski GD et al., J Cell Physiol 167(2), 1996). We included a discussion about it in the revised manuscript on line 314-320.

Reviewer 2**Comment 1:**

Studies in mouse and zebrafish have shown that embryonic and perinatal (larval and even adult fish) tenocytes possess progenitor properties in that they are able to proliferate. Hence, the authors rightfully aimed for the embryonic tenocytes with the goal to help repair tendon. It will be nice to exploit that property in this culture system. For example, what is the proliferative limitation of these in vitro iPSC-derived tenocytes/progenitors over time? Are there a mixture of progenitors and 'terminally differentiated tenocytes'? What are their relative proportions? While the scRNA-seq data give hints, a direct demonstration by cell biological data will be very helpful (e.g. using the markers they assigned for the stages/states of progression coupled with BrdU). These results may also help to explain the limited engraftment.

Answer to Comment 1:

To assess the proliferative characteristics of iPSC-tenocytes, we carried out EdU staining (co-stained with anti-MKX antibody) and cell growth curve analysis. 6.0% (positive rates to MKX) of iPSC-tenocytes were stained with EdU. Also, iPSC-tenocytes showed high proliferative properties over 8 days with a stable growth curve until reaching the confluence. Based on these results, it can be inferred that iPSC-tenocytes possess relatively high proliferative ability at least for 16 days from sclerotome stage. Regarding the cell maturity, we would consider iPSC-tenocytes as differentiated cells rather than progenitor cells because iPSC-tenocytes exhibited expression characteristics comparable to that of adult primary tenocytes (Fig. 3c). We created a new supplemental figure as Fig. 1d and Supplementary Fig. 1a and added explanation in the revised manuscript on line 116-120.

Comment 2:

While this study represents a leap forward for tenocyte development, the goal for tendon healing is a bit disappointing: At 4 weeks, there is no improvement over untreated (please change 'non-treat' to untreated, and 'non-injure' to uninjured) by a variety of measurements (Fig. 4), albeit a faster recovery at 2 weeks. Given the cell density increase and poor engraftment, one wonders where do the new cells come from? It was proposed, based on the secretome, that paracrine signaling by TGF β 3 and IGF1 is responsible. This does make sense based on previous studies. But, why not take this a step further by applying these two factors to injured tendons? I understand that the authors may wish to test this proposal in a subsequent study to elucidate their individual and combined roles in tendon healing extensively. However, as a proof of principle study, they should include the

positive effect of these two factors, even without knowing their target cell types – e.g. it could be the existing tenocytes, stem cells, or an unknown/unexpected cell type.

Answer to Comment 2:

Thanks for the suggestion, we corrected the words “non-treat” and “non-injure” to “untreated” and “uninjured” respectively. Regarding the therapeutic settings of TGFb3 and IGF1 in vivo, we investigated the contributions of both factors to the motor function recovery by injecting respective recombinant proteins after Achilles tendon rupture in rats. As shown in Fig. 6f, the AFI score of rats treated with IGF1 or TGFb3 was significantly restored at 2 weeks post-surgery compared to the control rats that were treated with PBS. Interestingly, the breaking stress of TGFb3 treated tendon indicated a recovery trend at 2 weeks, whereas the ultimate failure load did not show a significant difference after IGF1 nor TGFb3 administration (Supplementary Fig. 4e). We presume that exogenous IGF1 reduced functional deficit and accelerated motor function recovery via an anti-inflammatory mechanism as described in a previous work in an Achilles tendon injury rat model (C. Kurtz et al., The American Journal of Sports Medicine, 1999). With these data, we conclude that paracrine factors: IGF1 and TGFb3 contributed to the recovery of motor function after Achilles tendon rupture. We also assume that cell transplantation possesses a better therapeutic effect in comparison with the simple injection of IGF1 and TGFβ3 recombinant proteins. These data were newly added as Fig. 6f and Supplementary Fig. 4e, and explained in the manuscript on line 262-269, 283-290.

Reviewer 3

Comment 1:

While the protocol developed to obtain the iPSCs-tenocytes is undoubtedly robust, it is also complex, requiring several steps, relatively long times of culture and expensive reagents. How do the benefits of this approach compare to tenocytes derived/differentiated from stem cells sources that have been reported?

Answer to Comment 1:

We believe that the major benefits of our approach compare to the other studies that derive tenocytes from stem cells are robustness, simplicity, and xeno-free process of methods. For example, as Komura et al. reported in their innovative research in 2020 (ref20), they used embryoid bodies and empirically identified factors that transformed mouse iPSCs to tenocyte-like cells. The whole processes used in their study took 20 days and the FACS data showed that the differentiation efficiency was around 6% - 15%. In this regard, our xeno-free methods can differentiate human iPSCs-derived tenocytes in 19 days with 90%

efficiency. And our methods allow monolayer culture from the beginning. We have described some of points above in manuscripts as on line 310-314, however, it has not discussed the simplicity of our methods. Thus, we newly included such explanation on line 304-306.

Comment 2:

The authors mentioned the relevance of addressing the proposed cellular therapy in other tendons, namely in the rotator cuff, a major clinical problem. How do the authors envisage the application of a cellular therapy alone to solve injuries that are so demanding in terms of their biomechanical requirements?

Answer to Comment 2:

We envisage that cell therapy using the iPSC-tenocytes can be applied for a severe tendon injury possibly combined with the PRP treatments or growth factor treatments. However, to treat a severe rupture in the bone-tendon attachment domain, we assume that cellular therapy alone might not be applied, but can be combined with surgical procedures. We discussed it in the revised manuscript on line 331-333.

Comment 3:

Could authors also explain if, given that its well-known that the inflammatory response play a key role in tendon healing/regeneration, the response observed in the rat model selected could be different if implanted with rat cells instead of human derived cells?

Answer to Comment 3:

As you pointed out, the inflammatory response is the first phase of recovery after tendon injury and has an important role in promoting the regeneration. However, a persistent or excessive inflammatory response not only inhibits tendon regeneration, but also it can lead to further exacerbation of symptoms. In general, xenotransplantation without immunosuppressants would cause an excessive inflammatory reaction compared to allogeneic transplantation. Therefore, we guess that a relatively strong inflammatory response occurs in case of grafting human PSC-derived cells into rats like here in this study. Further investigations such as grafting of rat PSC-derived tenocytes and combined use of immunosuppressive agents would reveal the possibility for the better therapeutic effects of iPSC-tenocytes transplantation. We discussed it in the revised manuscript on line 333-338.

Reviewers' Comments:

Reviewer #1:

Remarks to the Author:

Overall, the authors did a nice job addressing this reviewer's initial concerns. In particular, the new data showing how the addition of functional data after the application of recombinant TGFb3 and IGF1 in vivo (Figure 6f) strengthens the manuscript.

However, there are a couple of concerns about interpretation of some of the new data.

The mechanics data is not interpreted or displayed correctly. This reviewer asked to have a graph showing the stress-strain response of the tendons (comment 4), not the force vs. displacement as shown in new Supplemental figure S3. These need to be plotted as stress (force normalized to cross sectional area) vs. strain (normalized deformation) to accurately demonstrate whether the material properties have been restored. In addition, these should be plotted on the same graph to facilitate comparison and it should be stated whether or not the tendons broke in the midsubstance or at the grips. If the tendons broke at the grips, then the "fracture" is an artifact of the grips and not the material properties of the tendon, and reporting the ultimate failure load is not useful (Figure 4e). Furthermore, "stress" and "load/force" cannot be used interchangeably (as done in lines 267, 285).

The accurate way to compare the material properties of the tendons is to compare the slope of the stress-strain curves at the same value of strain (or stress), along the linear portion of the J-shaped curve. For example, "the tangent modulus of the TGFb3-treated tendons at strain = 0.1 is higher/lower than the control." This reviewer will prefer that the tangent modulus is compared in Figure 4e, rather than ultimate breaking load; however, it is fine if the authors keep the current data as long as the curves in Figure S3 are updated to represent stress-strain rather than force-displacement.

In addition, the text describing the graphs needs to more accurately describe the true mechanical response of soft tissues. The repaired tendon does not have a "high viscosity." These tendons have an increased viscous contribution to the viscoelastic response, and are more ductile, compared to the controls.

It is appreciated that the authors added additional info to attempt to validate the hypothesis that there is more cartilage tissue by staining for Sox9 on sections of iPSC-sclerotome treated tendons (in response to comment 5); however, an adequate conclusion can only be made if both iPSC-sclerotome and iPSC-tenocyte treated tendons are shown. Sox9 may be upregulated in injured tendons, particularly proximal to the insertion at the calcaneus, and a quick look online suggests this might be the case: <http://www.ors.org/Transactions/61/0085.pdf>
The authors need to include the appropriate controls here.

Reviewer #2:

Remarks to the Author:

Nat Comm_278068_1 (revised)

This revised version contains new data and information. After reviewing my previous comments and the new data provided, I conclude that the authors have adequately address my previous comments.

I am somewhat disappointed that they did not perform combined treatment of IGF1 and TGFb3, given that they invested time and effort to test each individually for the revision. However, they did show tenocyte proliferation data (by these two factors individually), and did cite previous works regarding the known effect of IGF1 and TGFb3 in tendon development and treating tendon injury. Comparing to the cell transplantation data, it does seem reasonable to conclude that the transplanted cells provide more than these two factors. Using mesenchymal stem cells for

transplantation frequently leads to the same conclusion that they provide mainly paracrine 'support', rather than autonomous integration for repair. Nevertheless, this work represents a big step forward towards an alternative approach in facilitating tendon healing and motor function recovery.

Reviewer #3:

Remarks to the Author:

The authors have made significant efforts to improve the manuscripts and answered adequately to all the issues raised by the reviewers

Responses to the Reviewers' comments

Reviewer 1

Comment 1:

Overall, the authors did a nice job addressing this reviewer's initial concerns. In particular, the new data showing how the addition of functional data after the application of recombinant TGF β 3 and IGF1 in vivo (Figure 6f) strengthens the manuscript. However, there are a couple of concerns about interpretation of some of the new data.

The mechanics data is not interpreted or displayed correctly. This reviewer asked to have a graph showing the stress-strain response of the tendons (comment 4), not the force vs. displacement as shown in new Supplemental figure S3. These need to be plotted as stress (force normalized to cross sectional area) vs. strain (normalized deformation) to accurately demonstrate whether the material properties have been restored. In addition, these should be plotted on the same graph to facilitate comparison and it should be stated whether or not the tendons broke in the midsubstance or at the grips. If the tendons broke at the grips, then the "fracture" is an artifact of the grips and not the material properties of the tendon, and reporting the ultimate failure load is not useful (Figure 4e). Furthermore, "stress" and "load/force" cannot be used interchangeably (as done in lines 267, 285).

The accurate way to compare the material properties of the tendons is to compare the slope of the stress-strain curves at the same value of strain (or stress), along the linear portion of the J-shaped curve. For example, "the tangent modulus of the TGF β 3-treated tendons at strain = 0.1 is higher/lower than the control." This reviewer will prefer that the tangent modulus is compared in Figure 4e, rather than ultimate breaking load; however, it is fine if the authors keep the current data as long as the curves in Figure S3 are updated to represent stress-strain rather than force-displacement. In addition, the text describing the graphs needs to more accurately describe the true mechanical response of soft tissues. The repaired tendon does not have a "high viscosity." These tendons have an increased viscous contribution to the viscoelastic response, and are more ductile, compared to the controls.

Response to Comment 1:

We understand the reviewer's suggestions on the proper characterization of the material properties. Our understanding is that "strain" refers to the evaluation of the characteristics of the entirety of substances, and measuring "stress" is useful when comparing objects with different shapes. Considering the degeneration or shape-change (because 2–4 weeks had passed) of the Achilles tendon after injury, we agree with the reviewer's suggestion that it is important to evaluate properties of the entire regenerated tendon, which requires measurements of stress and strain. We have analyzed IGF1- or

TGFb3-treated tendons and have added a graph showing the stress-strain responses (Supplementary Fig. 4f). We have also compared the elastic moduli of the IGF-, TGFb3-, or PBS-treated tendons (Supplementary Fig. 4h). The elastic modulus was calculated based on the slope of a trendline drawn along the linear portion of the stress-strain curve (strain < 0.5). Consequently, TGFb3-treated tendons tended to have a slightly higher elastic modulus than the control tendons ($P = 0.08$), although the difference was not significant. Considering the data on AFI (Fig. 6f), ultimate failure load (Supplementary Fig. 4e), stress-strain curves (Supplementary Fig. 4f), breaking stress (Supplementary Fig. 4g), and elastic modulus (Supplementary Fig. 4h), we presume that exogenous growth factors reduced the functional deficit and accelerated motor function recovery via an anti-inflammatory mechanism. The mechanism by which extrinsic factors contribute to improved motility has previously been reported (Kurtz, C. A. et al., Am. J. Sports Med. 27, 363–369, 1999). We have explained these points on lines 271–274 and 293–297 in the revised manuscript.

Regarding the data on iPSC-tenocyte transplantation shown in Fig. 4, we regret that we do not have data on the surface area and length of tendons that are needed to calculate the stress and strain. To explain and display the data properly, we have revised the descriptions related to the recovery of material properties in our manuscript. We have now explained the data in the load-displacement curves as “In iPSC-tenocytes rats, the fracture load [N] and stiffness [N/mm] at the injured area were recovered compared to those in untreated rats” on lines 206–208. In addition, we have discussed limitations regarding the characterization of the material properties as “considering the degeneration or shape-change of the Achilles tendon after injury, a better way to characterize the material properties of the regenerated tendon after iPSC-tenocyte transplantation would be to generate stress vs. strain curves and compare the slopes at the same level of strain (or stress)” on lines 337–341. In addition, we have revised the explanation about the viscosity of the regenerated tendons on lines 203–206. In the revised manuscript, following your suggestion, we have explained this as “Untreated rats showed a gentle curve, which indicated increased ductility and a viscous contribution to the viscoelastic response, whereas iPSC-tenocytes rats and uninjured rats showed similar curves with a sharp drop immediately after the rupture”.

Furthermore, the force-displacement curves of iPSC-tenocyte-treated, untreated, and uninjured tendons have been plotted in the same graph (Supplementary Fig. 3a). Regarding the ultimate failure load test, regenerated Achilles tendons, which attach the gastrocnemius muscle to the calcaneus, were harvested, and then, the muscle segments were frozen in liquid nitrogen to prevent breakage at the grip domains. All regenerated

tendons used for that test broke in the midsubstance area. We have described this on line 500 in the revised manuscript.

Regarding the usage of the terms “stress” and “load/force” on lines 272 and 293, we have opted to keep the term “stress” since supplementary Fig. 4g shows the stress at the breaking point, not the load/force as shown in Fig. 4e and supplementary Fig. 4e.

Comment 2:

It is appreciated that the authors added additional info to attempt to validate the hypothesis that there is more cartilage tissue by staining for Sox9 on sections of iPSC-sclerotome treated tendons (in response to comment 5); however, an adequate conclusion can only be made if both iPSC-sclerotome and iPSC-tenocyte treated tendons are shown. Sox9 may be upregulated in injured tendons, particularly proximal to the insertion at the calcaneus, and a quick look online suggests this might be the case: <http://www.ors.org/Transactions/61/0085.pdf>. The authors need to include the appropriate controls here.

Response to Comment 2:

We have performed Sox9-immunostaining of the control sample, iPSC-tenocyte-treated tendon, which confirmed that there were no Sox9-positive cells around the transplanted area at 4 weeks after iPSC-tenocyte transplantation. As described in the article pointed by the reviewer, Sox9-lineage cells are recruited during the early phase of the tendon healing process. Considering that Sox9 is expressed in early tendon progenitors, but not in mature tenocytes, the regenerated tendon tissue at 4 weeks after iPSC-tenocyte transplantation can be presumed to have mature characteristics since no Sox9-positive cells were found. Therefore, we concluded that the Sox9-positive cells found in the iPSC-sclerotome are part of ectopic cartilage tissues. This is also consistent with the results of Alcian blue staining (Supplementary Fig. 4c). New staining data have been added in Supplementary Fig. 4d (left).

Reviewer 2

Comment 1:

This revised version contains new data and information. After reviewing my previous comments and the new data provided, I conclude that the authors have adequately address my previous comments.

I am somewhat disappointed that they did not perform combined treatment of IGF1 and TGFb3, given that they invested time and effort to test each individually for the revision. However, they did show tenocyte proliferation data (by these two factors individually), and

did cite previous works regarding the known effect of IGF1 and TGFb3 in tendon development and treating tendon injury. Comparing to the cell transplantation data, it does seem reasonable to conclude that the transplanted cells provide more than these two factors. Using mesenchymal stem cells for transplantation frequently leads to the same conclusion that they provide mainly paracrine 'support', rather than autonomous integration for repair. Nevertheless, this work represents a big step forward towards an alternative approach in facilitating tendon healing and motor function recovery.

Response to Comment 1:

As the reviewer pointed out, combined treatment with IGF1 and TGFb3 is expected to show better therapeutic effects on tendon healing and motor function recovery. We plan to address this point in a future study. We appreciate the reviewer's feedback.

Reviewer 3

Comment 1:

The authors have made significant efforts to improve the manuscripts and answered adequately to all the issues raised by the reviewers

Response to Comment 1:

We appreciate the time and effort the reviewer has dedicated to provide feedback on our paper.

Reviewers' Comments:

Reviewer #1:

Remarks to the Author:

The authors did a nice job addressing this reviewer's additional concerns and the new data/analyses is appreciated. One last request: Please change "elastic modulus" to "linear portion of the j-shaped curve" or "tangent modulus". While the term elastic modulus is commonly used in the biomechanics literature to describe the linear portion of the j-shaped curve, it is wrong to do so. The elastic modulus describes the resistance of material to deformation within the elastic (i.e. non-permanently deformed) range. When unloaded, the stress-strain response of the elastic material will follow the same path and all of the stored energy will be released. Tendons are viscoelastic materials and when deformed enough that a j-shaped curve is generated, the stress-strain curve will not follow the same path when the sample is unloaded and energy will be dissipated.

As stated in the previous review, a more accurate way to describe/compare the modulus is to measure the tangent of each sample at a given strain and compare that value (i.e. the tangent modulus at strain = XX). This reviewer is not asking for the moduli to all be recalculated for the same strain, just to make sure that the reader is aware that what is being reported is not the elastic modulus, rather the slope of the linear portion of the j-shaped curve.

Responses to the Reviewers' comments

Reviewer 1

Comment 1:

The authors did a nice job addressing this reviewer's additional concerns and the new data/analyses is appreciated. One last request: Please change "elastic modulus" to "linear portion of the j-shaped curve" or "tangent modulus". While the term elastic modulus is commonly used in the biomechanics literature to describe the linear portion of the j-shaped curve, it is wrong to do so. The elastic modulus describes the resistance of material to deformation within the elastic (i.e. non-permanently deformed) range. When unloaded, the stress-strain response of the elastic material will follow the same path and all of the stored energy will be released. Tendons are viscoelastic materials and when deformed enough that a j-shaped curve is generated, the stress-strain curve will not follow the same path when the sample is unloaded and energy will be dissipated.

As stated in the previous review, a more accurate way to describe/compare the modulus is to measure the tangent of each sample at a given strain and compare that value (i.e. the tangent modulus at strain = XX). This reviewer is not asking for the moduli to all be recalculated for the same strain, just to make sure that the reader is aware that what is being reported is not the elastic modulus, rather the slope of the linear portion of the j-shaped curve.

Response to Comment 1:

We changed the term of "elastic modulus" to "linear portion of the j-shaped curve" on line 271 and 291. We appreciate the time and effort the reviewer has dedicated to provide feedback on our paper.